# Structural basis of sequence-specific RNA recognition by the antiviral factor APOBEC3G

Hanjing Yang [1], Kyumin Kim [1], Shuxing Li[1,2], Josue Pacheco [1] & Xiaojiang S. Chen [1,2,3,4] ✉

An essential step in restricting HIV infectivity by the antiviral factor APOBEC3G is its incorporation into progeny virions via binding to HIV RNA. However, the mechanism of APOBEC3G capturing viral RNA is unknown. Here, we report crystal structures of a primate APOBEC3G bound to different types of RNAs, revealing that APOBEC3G specifically recognizes unpaired 5′-AA-3′ dinucleotides, and to a lesser extent, 5′-GA-3′ dinucleotides. APOBEC3G binds to the common 3′A in the AA/GA motifs using an aromatic/hydrophobic pocket in the non-catalytic domain. It binds to the 5′A or 5′G in the AA/GA motifs using an aromatic/hydrophobic groove conformed between the non-catalytic and catalytic domains. APOBEC3G RNA binding property is distinct from that of the HIV nucleocapsid protein recognizing unpaired guanosines. Our findings suggest that the sequence-specific RNA recognition is critical for APOBEC3G virion packaging and restricting HIV infectivity.

Human APOBEC3G (hA3G) is a member of apolipoprotein B mRNA editing enzyme catalytic polypeptide like (APOBEC) family[1–3], and is one of the most effective antiviral host restriction factors against exogenous retroviral pathogens such as human immunodeficiency virus type 1 (HIV-1) and endogenous retroviral derived remnants on the human genome[4–8]. hA3G is a DNA/RNA cytidine deaminase converting 3′ target cytidine (C) in 5′-CC-3′ or 5′-CCC-3′ motif to uridine (U) within single-stranded DNA (ssDNA)[9–15] or single-stranded RNA (ssRNA)[16–18]. Inside HIV-1 viral producer cells, a critical step before hA3G lashes antiviral activity against HIV-1 is its incorporation into progeny viral particles via binding HIV RNA[19–22]. Upon arriving at recipient cells, packaged hA3G deaminates HIV-1 negative cDNA during viral reverse transcription, causing massive hypermutation in the viral genome. In addition, hA3G shows a certain level of deaminase-independent HIV restriction activity[23–30], which is proposed to be through affecting HIV reverse transcriptase (RT) function or/and direct binding to viral RNA to interfere with viral cDNA replication.

hA3G has two cytidine deaminase domains in tandem, the N-terminal CD1 (or NTD) and C-terminal CD2 (or CTD). The ability to bind nucleic acids and form oligomerization is attributed mainly to CD1/NTD (referred to as CD1 hereafter), while deaminase activity is performed by CD2/CTD (referred to as CD2 hereafter). RNA cross-linking CLIP-seq data show that hA3G and two other human APOBEC family members, APOBEC3F (hA3F) and APOBEC3H (hA3H), display preferential binding to G-rich and A-rich RNA sequences that are not scanned by ribosomes during translation[31]. Multiple RNA binding residues of A3G have been identified previously by various approaches[32–41]. However, how A3G can selectively bind G-rich and A-rich RNA sequences to promote its virion packaging is unclear.

Poor solubility and the tendency to form large aggregates have been two major obstacles to A3G structural study. However, wild-type and soluble A3G variants of individual domains and soluble variants of the full-length A3G of human and primate origin yield multiple structures, revealing atomic arrangements of various forms of A3G. Specifically, there are three previously reported full-length structures of A3G variants containing no oligonucleotide (PDB ID: 6P40, 6P3X) or a deoxy-cytidine dinucleotide bound near the CD2 Zn-center (PDB ID: 6WMA)[41,42]. Despite some commonly shared interfaces at CD1-CD2

[1]Molecular and Computational Biology, Departments of Biological Sciences and Chemistry, Los Angeles, CA 90089, USA. [2]Center of Excellence in Nano-Biophysics, University of Southern California, Los Angeles, CA 90089, USA. [3]Genetic, Molecular and Cellular Biology Program, Keck School of Medicine, Los Angeles, CA 90033, USA. [4]Norris Comprehensive Cancer Center, University of Southern California, Los Angeles, CA 90033, USA. ✉e-mail: xiaojiac@usc.edu

boundaries, all these three reported forms show different CD1-CD2 domain arrangements with significant differences in CD1-CD2 bonding contacts. So far, the precise domain arrangement when rA3G is bound with RNA or DNA is unknown.

In this study, we obtained the co-crystal structures of a primate rhesus macaque APOBEC3G (rA3G) binding to several types of RNA substrates, resulting in monomeric rA3G-ssRNA and dimeric rA3G binding to 5′- or 3′-overhangs on both ends of a double-stranded RNA (dsRNA). Surprisingly, all these structures reveal a unique CD1-CD2 domain arrangement of rA3G when in complex with the RNA molecules. The structures also illustrate a molecular mechanism for rA3G to specifically recognize unpaired adenine dinucleotide (AA) RNA, and, to a lesser extent, GA (but not AG) dinucleotide RNA. This differs from the RNA binding property of the HIV nucleocapsid protein (NCp7), which targets unpaired guanosines[43–47]. Our structure-guided mutational studies using both rA3G and hA3G suggest that the sequence-specific RNA-recognition mechanism is critical for A3G virion packaging and restriction against HIV infectivity.

## Results

### RNA binding property of rhesus macaque A3G (rA3G)

We tried to obtain the complex structure of A3G binding to RNA to understand how A3G interacts with RNA and how such RNA interactions mediate oligomerization and virion packaging. We successfully produced a soluble variant of rA3G carrying a replacement of N-terminal domain loop 8 ($_{139}$CQKRDGPH$_{146}$ to $_{139}$AEAG$_{142}$)[41] and designated this construct as rA3G$_{R8}$ in this study. We found that rA3G$_{R8}$ expressed in *Escherichia coli* binds to bacterial RNA that can be degraded to various degrees by RNase treatment and can be partially dissociated from rA3G$_{R8}$ during purification[41]. As a result, fractions of various amounts of RNA-bound or RNA-free rA3G$_{R8}$ can be purified. An attempt to crystallize the purified rA3G$_{R8}$ fraction with bound RNA was unsuccessful, possibly due to heterogeneous species of co-purified RNA.

We decided to reconstitute rA3G$_{R8}$-RNA complexes in vitro using the purified RNA-free fraction of rA3G$_{R8}$ and short synthetic RNA. Preliminary RNA binding tests using an RNA molecule comprised of a dsRNA with both 5′ and 3′ overhangs yielded a pronounced peak shift by size-exclusion chromatography (SEC). As a result, three types of RNA were designed to evaluate sequence-specific RNA binding: dsRNA with a short 5′ or 3′ overhang (RNA1 or RNA2, Fig. 1a) and ssRNA (RNA3, Fig. 1a). Two variable bases NN (NN = GG, AA, CC, or UU) were introduced to the short 5′ and 3′ overhangs at the dsRNA junction, and in the middle of the short ssRNA sequences. Formation of the annealed dsRNA with overhangs was verified by native gel electrophoresis (Supplementary Fig. 1a, b). The dissociation constant $K_D$ of twelve RNA molecules was measured and ranked using electrophoretic mobility shift assay (EMSA, Fig. 1a). Surprisingly, the three best binders were adenine dinucleotides (AA) in all three types of RNA: the 5′ overhang dsRNA (RNA1-AA), 3′ overhang dsRNA (RNA2-AA), and ssRNA (RNA3-AA) with $K_D$ of 10 to 17 nM. Discrete bands of rA3G$_{R8}$ bound to the AA dinucleotide RNA were visible with RNA1-AA and RNA2-AA at rA3G$_{R8}$ concentrations between ~10 to ~100 nM, and large aggregates appeared at rA3G$_{R8}$ concentrations ~300 nM or higher. The RNA containing guanine dinucleotides (GG) at 5′ and 3′ overhang (RNA1-GG and RNA2-GG) showed $K_D$ ranging from 170 to 250 nM with more than tenfold less affinity. The RNA containing pyrimidine dinucleotides CC or UU at 5′ overhang or 3′ overhang (RNA1-CC/UU and RNA2-CC/UU) showed even less affinity, with $K_D$ values around 400 nM. The least favored were ssRNA with pyrimidine dinucleotide sequences (RNA3-CC and RNA3-UU). The ssRNA also displayed the most dramatic difference in $K_D$ value between AA and GG/CC/UU sequences. Collectively, these results indicate that rA3G has a strong preference for unpaired AA dinucleotides in RNA.

Replacement of the 5′-AA-3′ sequence to 5′-GA/CA/UA-3′ or 5′-AG/AC/AU-3′ reduced rA3G$_{R8}$ binding to ssRNA (Fig. 1b). These results

confirm that rA3G$_{R8}$ prefers the sequence motif AA, followed by the sequence motif GA with $K_D$ of 47 nM. The rest of the sequence motifs UA, AU, AG, CA, and AC show reduced affinity, but all bind better than GG, CC, and UU. Of note, rA3G$_{R8}$ has a bias towards GA over AG, which is discussed in the later sections.

RNA binding property of rA3G$_{R8}$ was compared to that of the HIV nucleocapsid protein (NCp7), a nucleic acid chaperone that prefers unpaired guanosines (Gs) in RNA[43–47] and is critical for the highly selective HIV RNA packaging process[48,49]. A purified RNA-free fraction of the NCp7 protein tagged by an N-terminal glutathione *S*-transferase (GST-NCp7, Supplementary Fig. 2a) was tested with the initial set of 12 RNA molecules containing the two variable bases NN using EMSA. As expected, the three best binders for GST-NCp7 were guanine dinucleotides (GG) in the 5′ overhang dsRNA and 3′ overhang dsRNA (RNA1-GG and RNA2-GG, Supplementary Fig. 2b), and ssRNA (RNA3-GG, Fig. 1c). Further tests using ssRNA containing GA or AG confirm that GST-NCp7 prefers unpaired Gs with the ranking GG > GA/AG > AA/CC/UU (Fig. 1c), which is distinct from the RNA binding property of rA3G$_{R8}$ as AA > GA > AG > GG > CC/UU (Fig. 1a, b).

### Overall architecture of rA3G-RNA complexes with AA dinucleotide RNA

We confirmed that rA3G$_{R8}$ or rA3G$_{R8/K128D}$ (containing a K128D mutation, which mimics human A3G at position 128) prefers AA dinucleotide RNA by analytical size-exclusion chromatography (SEC) (Supplementary Fig. 3). Complexes of rA3G with three different AA dinucleotide RNAs were eluted ahead of the protein alone or RNA alone, while no peak shift was detected when mixing rA3G with the other nine RNA molecules. These results indicate that rA3G forms stable complexes with unpaired AA dinucleotide-containing RNA under the assay condition.

We proceeded to co-crystallize rA3G with the three different types of AA dinucleotide-containing RNAs. RNA1-AA and RNA2-AA were modified to potentially facilitate crystal packing (Fig. 2a, d). In both RNA molecules, the length of dsRNA was shortened from 11 bp (dsRNA$_{11}$) to 8 bp (dsRNA$_8$). Additionally, RNA1-AA was modified to 5′ U$_1$U$_2$A$_3$A$_4$-dsRNA$_8$-C$_{13}$C$_{14}$U$_{15}$U$_{16}$U$_{17}$U$_{18}$ with 5′ and 3′ overhangs on both strands (designated as RNA1-AA$_{xtal}$, with dsRNA$_8$ 5′-CGCUGCGG-3′/5′-CCGCAGCG-3′; Fig. 2a). RNA2-AA was modified with 3′ overhang on both strands (5′ dsRNA$_8$-A$_9$A$_{10}$U$_{11}$U$_{12}$U$_{13}$U$_{14}$, designated as RNA2-AA$_{xtal}$, with dsRNA$_8$ 5′-CCCGUGGG-3′/5′-CCCACGGG-3′; Fig. 2d). RNA3-AA was used as is without further modification (5′ U$_1$U$_2$U$_3$U$_4$A$_5$A$_6$U$_7$U$_8$U$_9$U$_{10}$, also designated as RNA3-AA$_{xtal}$, Fig. 2g). rA3G$_{R8}$ carrying a catalytically inactive mutation E259A (with K128 or D128 at position 128) was used in the co-crystallization study (designated as rA3G$_{R8/E259A}$). After multiple attempts, diffracting crystals of three rA3G$_{R8/E259A}$-RNA complexes were obtained, and their structures were determined (Table 1).

Complexes with RNA1-AA$_{xtal}$ and RNA2-AA$_{xtal}$ were determined as two different types of dimers with two rA3G molecules binding to one RNA molecule at 2.90 and 3.10 Å, respectively (Fig. 2a, b, d, e and Table 1). In both cases, the electron density maps of RNA were of sufficient quality to allow de novo building of a complete dsRNA$_8$ plus three nucleotides (3-nt) on each AA dinucleotide-containing overhang (5′-U$_2$A$_3$A$_4$-dsRNA$_8$-3′ and 5′-dsRNA$_8$-A$_9$A$_{10}$U$_{11}$-3′, Fig. 2c, f). In both dimer models, there are no direct contacts between the two rA3G$_{R8/E259A}$ monomers (Fig. 2b, e), which is reminiscent of A3H-dsRNA complexes showing RNA-assisted dimer formation[50–52]. Most of the protein-RNA contacts were mapped to the CD1 domain (Fig. 2b, e), which is consistent with the literature that CD1 is essential for RNA association[38–41,53]. The structures also reveal the participation of CD2 in RNA binding, which is further discussed in the later sections.

Each rA3G$_{R8/E259A}$ molecule has strong interactions with the 3-nt AA dinucleotide overhang (Fig. 2a, b, d, e). Averaged RNA surface area

buried with one rA3G$_{R8/E259A}$ is ~820 Å$^2$ in RNA1-AA$_{xtal}$ and ~841 Å$^2$ in RNA2-AA$_{xtal}$. With two rA3G$_{R8/E259A}$ monomers contacting one RNA, approximately 1640 to 1680 Å$^2$ of the RNA surface area centered around the AA dinucleotides was buried in a rA3G-RNA dimer, which accounts for roughly 28% of the total RNA surface area, leaving most of the dsRNA surface exposed. This contrasts with A3H-dsRNA containing 7-bp dsRNA with 2-nt overhang, where over 40% of the RNA surface area is buried (~2180 Å$^2$)[50]. Lacking sufficient protection to RNA bound with rA3G is consistent with observations that rA3G-RNA is more

sensitive to RNase treatment than A3H-dsRNA during recombinant protein purification.

The third complex of rA3G with RNA3-AA$_{xtal}$ was determined as a monomer with one rA3G$_{R8/E259A}$ molecule binding to one ssRNA molecule at 2.10 Å (Fig. 2g, h and Table 1). The electron density of RNA was sufficiently featured for building 6-nt: 5′-U$_4$A$_5$A$_6$U$_7$U$_8$U$_9$-3′ out of the 10-nt 5′-U$_1$U$_2$U$_3$U$_4$A$_5$A$_6$U$_7$U$_8$U$_9$U$_{10}$-3′ (Fig. 2i). About 47% of the resolved 6-nt ssRNA surface area is buried, with most of the rA3G-ssRNA contacts placed around AA dinucleotides and on the CD1

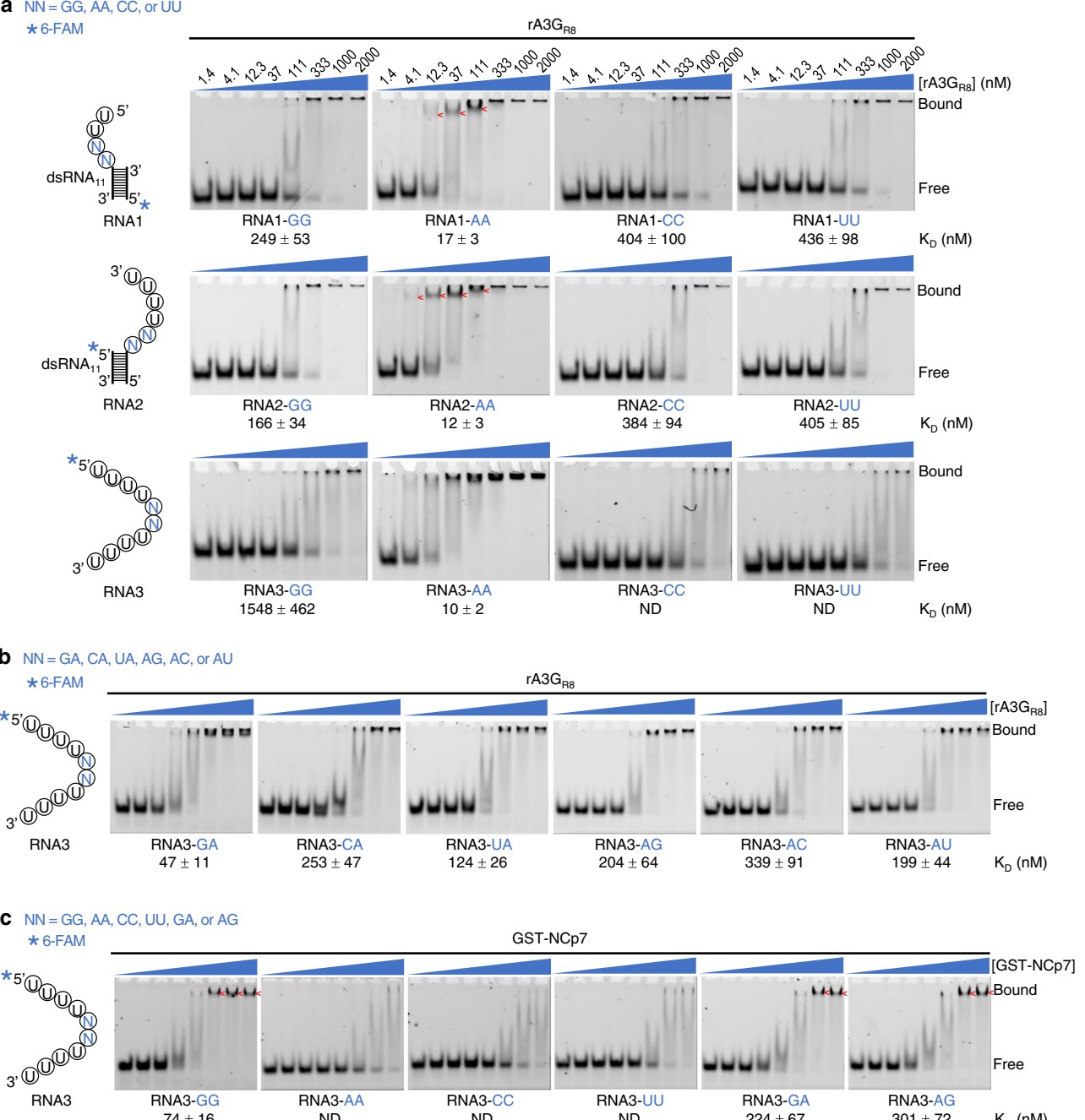

**Fig. 1 | rA3G$_{R8}$ and GST-NCp7 binding to various synthesized RNA molecules visualized by electrophoretic mobility shift assay (EMSA). a** rA3G$_{R8}$ binding to three types of RNA that contain two variable bases NN: 5′ overhang dsRNA (RNA1), 3′ overhang dsRNA (RNA2), and ssRNA (RNA3). NN = GG, AA, CC, or UU. dsRNA$_{11}$ represents an 11-bp double-stranded RNA molecule (Supplementary Table 1). **b** rA3G$_{R8}$ binding to RNA3-NN, where NN = GA, CA, UA, AG, AC, or AU. **c** GST-NCp7 binding to RNA3-NN, where NN = GG, AA, CC, UU, GA, or AG. RNA at a fixed concentration of 10 nM was incubated with rA3G$_{R8}$ or GST-NCp7 at various concentrations (1.4, 4.1, 12.3, 37, 111, 333, 1000, and 2000 nM). The estimated $K_D$ values as mean values ± SD are listed below individual representative EMSA gel images. $n = 3$ independent experiments. The asterisk marks the location of 6-FAM. Discrete-shifted species are marked with red arrows. ND not determined. Source data are provided as a Source Data file.

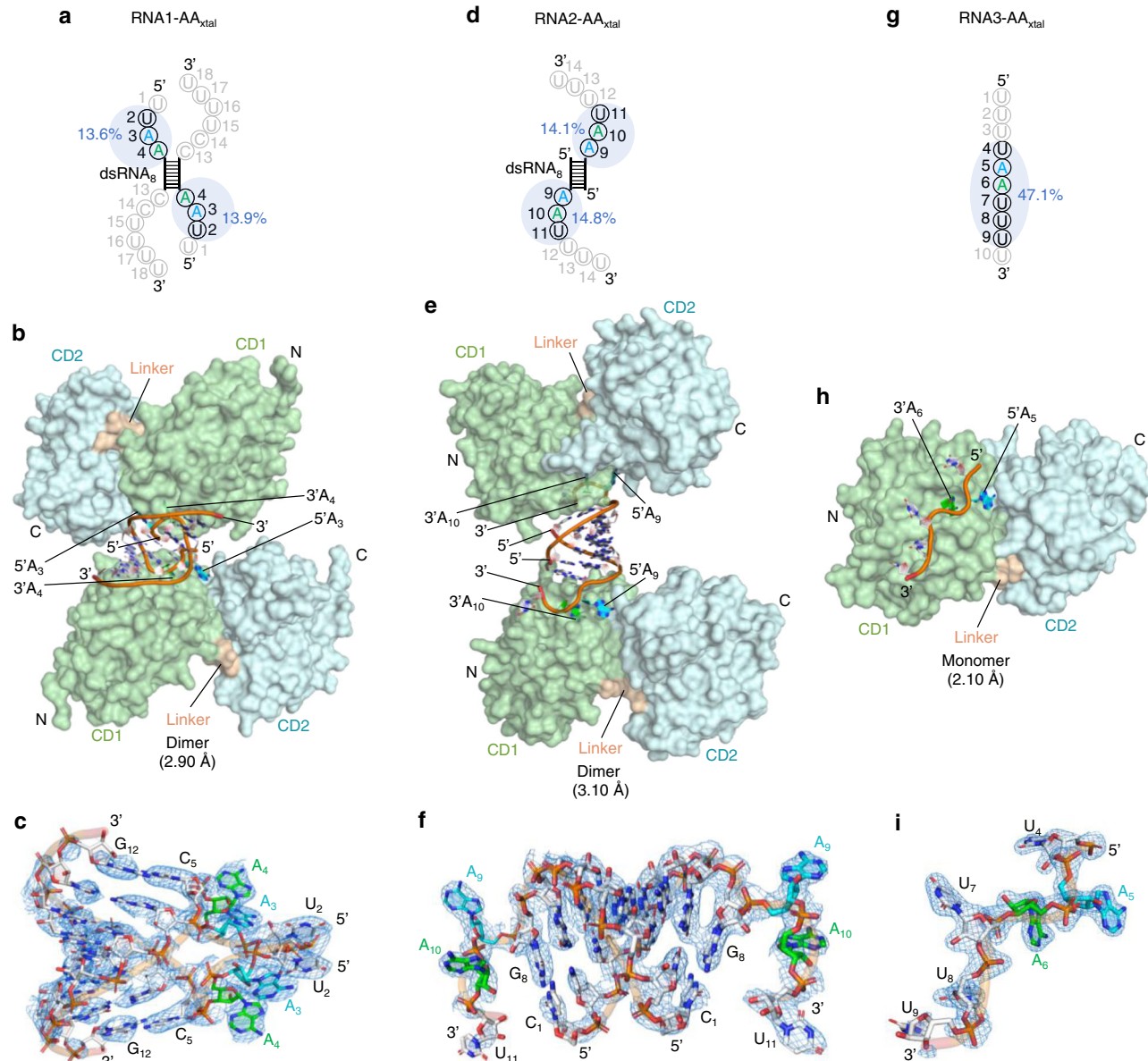

**Fig. 2 | Overall architecture of rA3G$_{R8/E259A}$ in complex with three types of AA-containing RNA.** To distinguish the two adenosines in the sequence motif 5′-AA-3′, we designated the 5′-end A as 5′A (in cyan) and the 3′-end A as 3′A (in green). **a** A schematic of the 5′-overhang AA dinucleotide dsRNA (RNA1-AA$_{xtal}$). **b** The co-crystal structure of rA3G$_{R8/E259A}$ and RNA1-AA$_{xtal}$ as a dimer (PDB ID: 7UU5). **c** A 2Fo-Fc electron density map of the resolved RNA1-AA$_{xtal}$ contoured at 1 σ level. **d** A schematic of the 3′-overhang AA dinucleotide dsRNA (RNA2-AA$_{xtal}$). **e** The co-crystal structure of rA3G$_{R8/E259A}$ and RNA2-AA$_{xtal}$ as a dimer (PDB ID: 7UU3). **f** A 2Fo-Fc electron density map of the resolved RNA2-AA$_{xtal}$ contoured at 1.5 σ level.

**g** A schematic of the ssRNA (RNA3-AA$_{xtal}$). **h** The co-crystal structure of rA3G$_{R8/E259A}$ and RNA3-AA$_{xtal}$ as a monomer (PDB ID: 7UU4). **i** A 2Fo-Fc electron density map of the resolved RNA3-AA$_{xtal}$ contoured at 1.5 σ level. In panels **a**, **d**, **g**, the RNA bases not marked in light gray are resolved in the structures. dsRNA$_8$ represents an 8-bp double-stranded RNA molecule (Supplementary Table 1). The percentage of the buried RNA surface is shaded in light blue with the percentage value labeled. In panels **b**, **e**, and **h**, CD1 (NTD) is colored in pale green, CD2 (CTD) in pale cyan, and the CD1-CD2 linker $_{194}$RHL$_{196}$ in wheat. See Supplementary Fig. 10 for additional views of the dimers shown in panels **b** and **e**.

domain (Fig. 2g, h), which is consistent with the findings from the two rA3G$_{R8/E259A}$-dsRNA complexes described above.

## Domain arrangement of CD1 and CD2

Even though the three RNA-bound rA3G$_{R8/E259A}$ structures have different crystal packing, the individual rA3G$_{R8/E259A}$ monomer structures from all three RNA-bound complex structures align well with each other (Supplementary Fig. 4a), with root-mean-square deviations (rmsds) of 0.52−0.66 Å over ~2390−2541 pairs of atoms. This indicates the structures are essentially identical despite binding to three different types of RNA. A significant difference in the CD1-CD2 domain arrangement of A3G was observed between the RNA-bound

conformation and the three available structures published earlier[41,42]. If aligning the CD1 domains of the full-length A3G in the RNA-bound form with the three previously reported full-length structures of A3G variants, the CD2 domain from the RNA-bound rA3G has approximately +135° turn and +180° rotation from the CD2 of the two apo-rA3G forms (PDB ID: 6P40, 6P3X) (Fig. 3a) and +90° turn and −45° rotation from the CD2 of the hA3G form (PDB ID: 6WMA) (Supplementary Fig. 4b, c). The above comparison of the domain arrangement reveals that a flexible CD1-CD2 linker $_{194}$RHL$_{196}$ may allow interchange between a few discreet low-energy states of CD1-CD2 orientations for the apoproteins but permit only one conformation upon RNA binding. For the three reported conformations, though,

**Table 1 | Crystallographic data collection and refinement statistics**

| PDB ID | 7UU5 rA3G/RNA1-AA$_{xtal}$ (5' Overhang dsRNA) | 7UU3 rA3G/RNA2-AA$_{xtal}$ (3' Overhang dsRNA) | 7UU4 rA3G/RNA3-AA$_{xtal}$ (ssRNA with AA) | 8EDJ rA3G/RNA3-GA$_{xtal}$ (ssRNA with GA) |
|---|---|---|---|---|
| **Data collection** | | | | |
| Space group | P1 | P1 | P2$_1$2$_1$2$_1$ | P2$_1$2$_1$2$_1$ |
| *Cell dimensions* | | | | |
| a, b, c (Å) | 61.3, 71.6, 72.7 | 60.7, 67.2, 79.6 | 54.2, 68.0, 126.8 | 55.3, 68.1, 129.3 |
| α, β, γ (°) | 119.4, 111.0, 90.1 | 98.3, 108.7, 112.7 | 90, 90, 90 | 90, 90, 90 |
| Resolution (Å) | 46.5-2.90 | 45.7-3.10 | 46.4-2.10 | 40.7-1.83 |
| | (3.00-2.90)$^a$ | (3.21-3.10)$^a$ | (2.18-2.10)$^a$ | (1.90-1.83)$^a$ |
| $R_{merge}$ | 0.107 (0.503) | 0.130 (0.763) | 0.114 (0.621) | 0.085 (0.935) |
| CC1/2 | 0.994 (0.845) | 0.990 (0.557) | 0.998 (0.922) | 0.999 (0.953) |
| $I/\sigma I$ | 11.5 (1.6) | 8.9 (2.5) | 14.9 (3.1) | 21.0 (3.9) |
| Completeness | 0.98 (0.92) | 0.98 (0.94) | 0.98 (0.87) | 1.00 (1.00) |
| Redundancy | 3.3 (3.3) | 3.3 (2.4) | 12.9 (10.6) | 12.7 (13.2) |
| **Refinement** | | | | |
| Resolution (Å) | 46.5-2.90 | 45.7-3.10 | 46.4-2.10 | 40.7-1.83 |
| | (3.00-2.90)$^a$ | (3.21-3.10)$^a$ | (2.18-2.10)$^a$ | (1.90-1.83)$^a$ |
| $R_{work}/R_{free}$ | 0.189/0.241 | 0.170/0.210 | 0.184/0.205 | 0.169/0.192 |
| No. of atoms | 6771 | 6627 | 3466 | 3566 |
| Macromolecules | 6767 | 6623 | 3228 | 3229 |
| Ligand/ion | 4 | 4 | 7 | 7 |
| Water | | | 231 | 330 |
| B-factor | | | | |
| Macromolecules | 86.9 | 94.3 | 45.2 | 37.3 |
| Ligand/ion | 107.9 | 84.0 | 40.5 | 27.6 |
| Water | | | 45.0 | 44.7 |
| *R. m. s. deviations* | | | | |
| Bond lengths (Å) | 0.004 | 0.004 | 0.004 | 0.006 |
| Bond angles (°) | 1.00 | 0.97 | 0.70 | 0.80 |

$^a$Highest-resolution shell is shown in parentheses.

individual mutations carried in these soluble variants may affect the CD1-CD2 domain arrangement, and the effect of crystal packing could not be excluded.

The CD1-CD2 orientation locked in the three RNA-bound structures results in a distinct CD1-CD2 interface, which involves interactions mainly between h5/h6 on CD1 and β2/lp3/h2 on CD2 with a buried area of ~720 Å² that is comparable to the buried interface area in the different apo forms. Surprisingly, the interface identified herein overlaps with the previously described CD1 h6-h6 apo-rA3G dimer interface[41], thus, the RNA-bound form precludes the h6-h6 mediated dimerization seen in the apo form. About 12 amino acid residues from CD1, two residues from the linker region, and 17 residues from CD2 are mapped to the interface (Fig. 3b and Supplementary Fig. 5).

**The core structure of rA3G bound to the AA dinucleotides**
The arrangement of CD1 and CD2 in the RNA-bound structures bridges a positively charged electrostatic potential surface area near the zinc-catalytic center of CD2 and a largely positive charged surface area of CD1 (Fig. 3c). Comparing the three rA3G-RNA structures reveals a conserved AA dinucleotide interaction core structure within the sequence motif 5′-(U/G)AA-3′ with rmsds ranging from 0.130 for 42 pairs of atoms to 0.259 for 58 pairs of atoms. The directionality of the core sequence is with its 5′-end pointing upward towards the side where locates the loop 10–h6 of CD1 near the zinc-catalytic center of CD2 (Fig. 3b, c). The crystal structure of rA3G$_{R8/E259A}$-ssRNA (resolution 2.10 Å) is used here as an example to depict the detailed interactions between rA3G and the AA (or A$_5$A$_6$) dinucleotide in the 5′-U$_4$A$_5$A$_6$-3′ core sequence (Fig. 4). To distinguish the two adenosines in the

sequence motif 5′-AA-3′, we designated the 5′-end A as 5′A (or 5′A$_5$) and the 3′-end A as 3′A (or 3′A$_6$) hereafter.

3′A$_6$ in the AA (or A$_5$A$_6$) dinucleotide is bound to an aromatic/hydrophobic "cave", a pocket outside but right next to the zinc-center of CD1, forming strong hydrophobic interactions with the solvation energy effect of −2.47 kcal/mol (Fig. 4). This pocket is largely pre-formed and comprised of mainly hydrophobic and aromatic residues: $_{25}$PILS$_{28}$ from loop 1, Y59 from loop 3, W94 from loop 5, and $_{123}$LYYFW$_{127}$ from loop 7 (Fig. 4). Inside the pocket, there are four hydrogen bonds formed (Fig. 4d, bond lengths 2.8–3.1 Å) between N1, N6, and N7 nitrogen atoms of 3′A$_6$ and the carbonyl and amide groups within CD1-loop 1 (P25, L27, along the Hoogsteen edge of adenine) and loop 7 (L123, Y125, along the Watson–Crick edge of adenine), fitting 3′A$_6$ base snugly into the pocket. Furthermore, OP1 of U$_7$ phosphate oxygen (at the 3'-end of 3′A$_6$) forms hydrogen bonds with the side chains of three tyrosine residues Y59, Y124, and Y125 (bond lengths 2.5, 3.4, and 2.6 Å), indicating that 3′-phosphate is critical in RNA binding, and three tyrosine residues have somewhat overlapping RNA binding function. In silico analysis reveals that placing a guanine nucleotide inside the pocket would generate a clash between the C-2 amino group of the guanosine and the aromatic ring of Y125. Additionally, the C-6 carbonyl group of the guanine base is highly unfavorable when in proximity (3.0–3.1 Å) to the two backbone carbonyl groups of the residues P25 and L123 (Supplementary Fig. 6). Therefore, the pocket is likely adenosine specific for the 3′A$_6$ in the AA dinucleotides.

5′A$_5$ in the AA (or A$_5$A$_6$) dinucleotide is bound to a groove conformed by the CD1 and CD2, also forming strong hydrophobic interactions with the solvation energy effect of −3.09 kcal/mol (Fig. 4a–c).

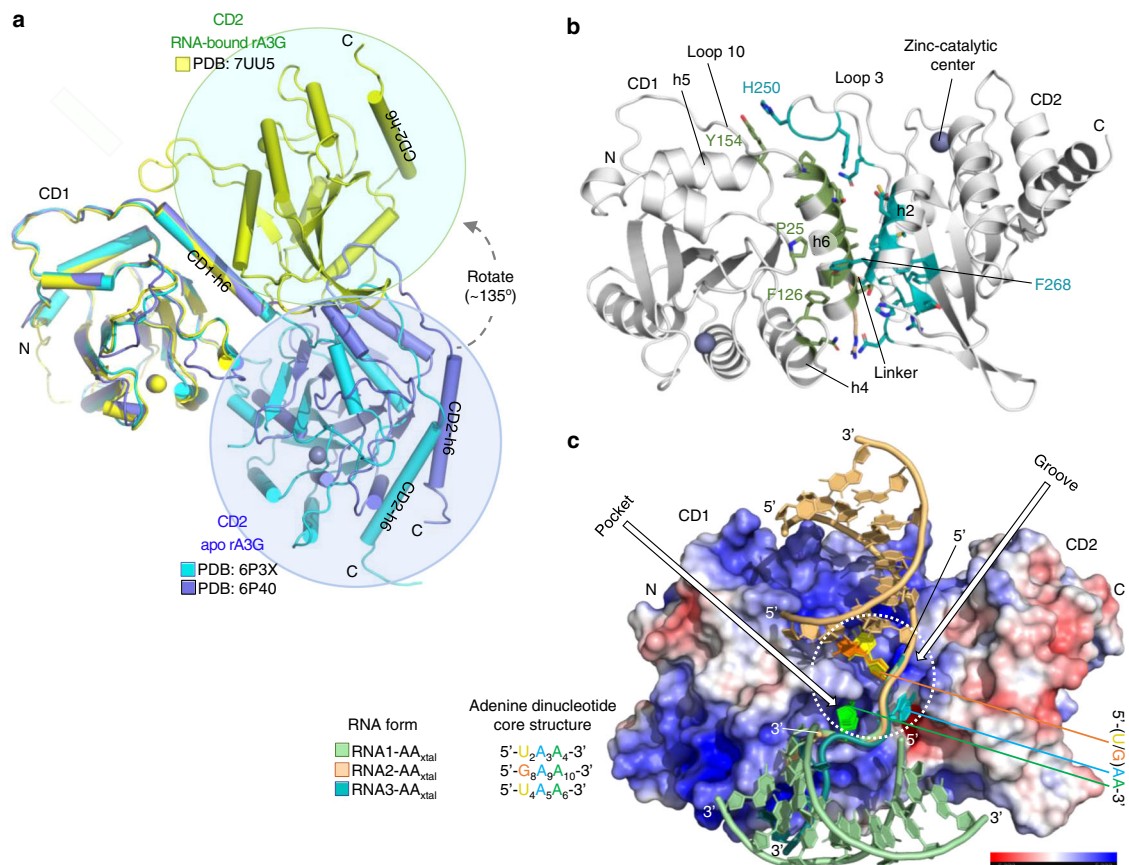

**Fig. 3 | rA3G CD1-CD2 domain arrangement and the conserved core structure of rA3G binding to the AA dinucleotides. a** Superimposition of the CD1 of the RNA-bound rA3G structure and two apo-rA3G structures, showing the CD2 turns ~135° and rotates about 180° from the CD2 of apo-rA3G. **b** The interface between CD1 and CD2 in the RNA-bound rA3G$_{R8/E259A}$ structure (PDB ID: 7UU4). The interface residues with a buried surface area larger than 10 Å$^2$ are shown. See Supplementary Fig. 5 for the detailed arrangement of the interface residues. **c** The surface electrostatic potential of rA3G$_{R8/E259A}$ with super-imposition of the three bound RNA molecules. The conserved core-binding structure of the sequence motif 5′-(U/G)AA-3′ (inside a white dotted circle) is indicated. The two structural features on the rA3G$_{R8/E259A}$ surface, the pocket and the groove, are indicated. The electrostatic surface potential is colored from red to blue with −5 to +5 kT/e. The molecules in panels **b** and **c** are positioned similarly.

It is surrounded by five hydrophobic and aromatic residues with three residues, I26 and $_{126}$FW$_{127}$ shared between A$_5$ and A$_6$, and two CD2 residues, F268 and K270, unique for A$_5$. One hydrogen bond is formed between N6 nitrogen atom of 5′A$_5$ and the main-chain carbonyl group of F268 (bond length 2.8 Å). K270 of CD2 makes contacts through hydrophobic packing with 5′A$_5$ via its aliphatic side chain. Unlike the tight interaction between the cave-like pocket and the 3′A$_6$, the shape of this groove could snugly fit with a purine base (A$_5$ or G$_5$) with similar hydrophobic stacking interaction, but a much weaker fit with a smaller pyrimidine base (U$_5$ or C$_5$). However, a carbonyl group (C-6) of guanine (G$_5$) at the equivalent position is unfavorable next to the carbonyl group of F268. This could explain the observed rA3G$_{R8}$ binding preference for the sequence motif 5′-A$_5$A$_6$–3′ over 5′-G$_5$A$_6$–3′ (Fig. 1a, b). To verify this hypothesis, the crystal structure of rA3G bound to GA RNA will be described in the following section (Fig. 5), with further discussion of the RNA binding property of this groove.

U$_4$ at the 5′-end of the AA dinucleotide core structure is held between R24 and S28 on the rA3G surface, suggesting a none-base specific binding with the solvation energy effect of 3.05 kcal/mol (Fig. 4a, b). R24 not only forms pi-stacking with U$_4$ base but also interacts with the phosphate backbone of the RNA, forming two hydrogen bonds with OP2 of the U$_4$ 5′-phosphate oxygen (bond lengths 3.0 and 3.3 Å). S28 uses its side chain and the main chain to form two hydrogen bonds with U$_4$ 2′-OH present only in RNA (bond lengths 2.8 and 3.1 Å), suggesting that S28 may specify RNA

interaction. The S28 side chain and main chain also form two hydrogen bonds with U$_4$ sugar (O3′) and base (O2) (bond lengths 3.5 and 3.1 Å). These RNA-specific interactions of R24 and S28 are consistent with the published mutational data, showing R24A and S28A reduced RNA binding[33,34]. I26 and L27 provide hydrophobic contacts with U$_4$. Additionally, the NH2 group of K270 is 4.7 Å away from the U$_4$ phosphate backbone, which has the potential to form a weak hydrogen bond interaction directly or through a water molecule.

In summary, our structural and biochemical evidence demonstrate that rA3G specifically binds to the 5′-A$_5$A$_6$-3′ dinucleotides within the sequence 5′-U$_1$U$_2$U$_3$U$_4$A$_5$A$_6$U$_7$U$_8$U$_9$U$_{10}$-3′, forming exceptionally strong hydrophobic interactions via the two adjacent surface features: the pocket on CD1 (for 3′A$_6$) and the CD1-CD2 interface groove (for 5′A$_5$). Furthermore, 14 hydrogen bonds form between the sequence motif 5′-U$_4$A$_5$A$_6$-3′ and the rA3G residues located on the CD1 loops 1, 3, 7, and CD2 (F268). Lastly, rA3G binds to the rest of the ssRNA (U$_7$/U$_8$/U$_9$), forming additional seven hydrogen bonds (Supplementary Fig. 7a).

## rA3G bound to GA dinucleotide RNA

To further gain structural basis into rA3G$_{R8}$ binding preference of the purine dinucleotide as AA > GA > AG > GG (Fig. 1a, b), co-crystallization trials were set up in parallel with ssRNA (RNA3) containing the sequence motifs GA, AG, and GG, with RNA3-AA (5′-U$_1$U$_2$U$_3$U$_4$A$_5$A$_6$U$_7$U$_8$U$_9$U$_{10}$-3′) as a positive control. Under similar crystallization

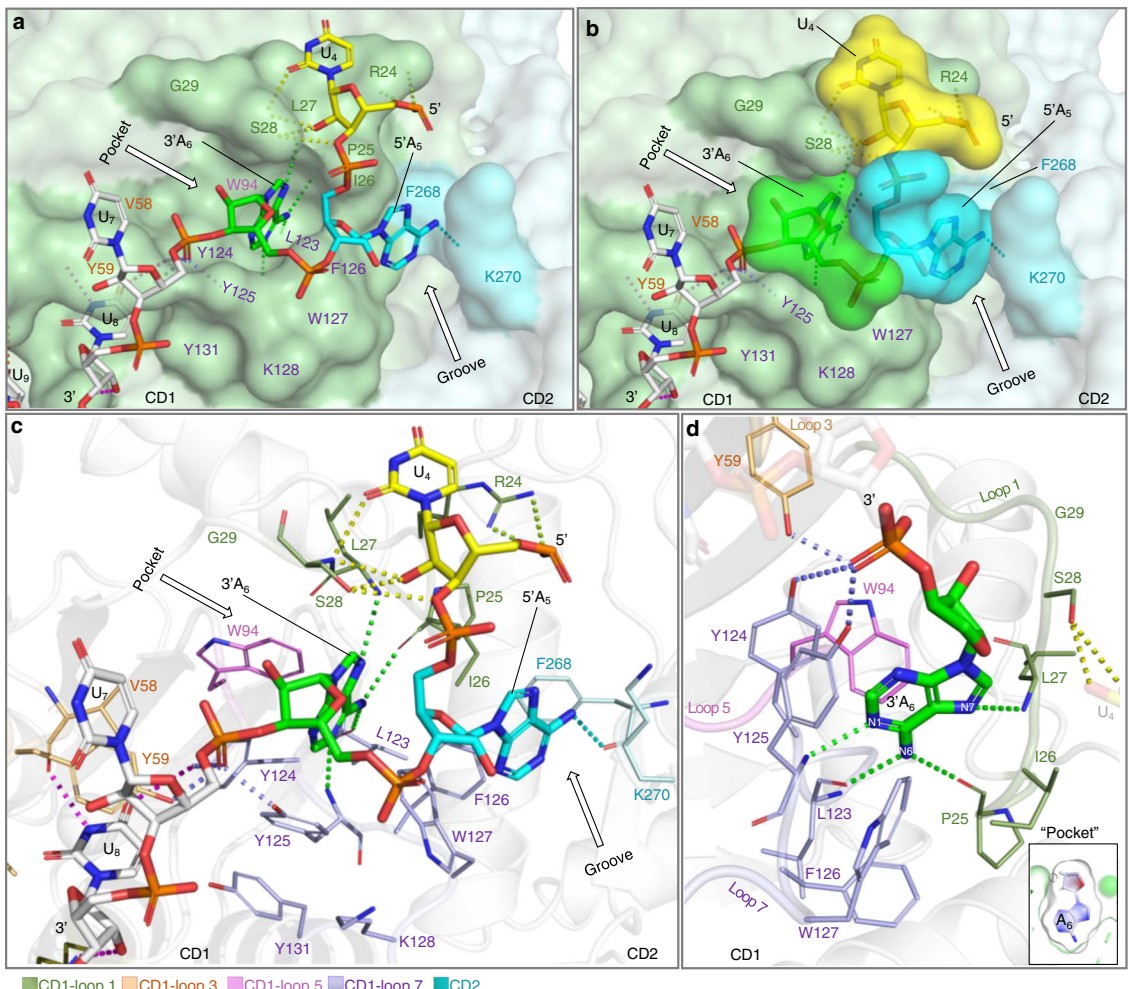

**Fig. 4 | Detailed interactions between rA3G and the core sequence motif 5′-U₄A₅A₆-3′ in the ssRNA (RNA3-AA_xtal) co-crystal structure (PDB ID: 7UU4).** The color key of the amino acid residues is below panel **c**. **a** Stick and **b** surface representations of the RNA sequence motif 5′-U₄A₅A₆-3′, showing 3′A₆ (in green) inside the pocket, 5′A₅ (in cyan) at the interface groove, and 5′-end U₄ (in yellow) held between R24 and S28 on the rA3G surface. The two surface features: the pocket and the groove, are indicated. Dashes: hydrogen bonds. **c** Stick representation of rA3G residues and the core sequence motif 5′-U₄A₅A₆-3′, showing 14 hydrogen bonds: six with U₄, one with 5′A₅, four with 3′A₆, and three with the phosphase group at the 3′-end of 3′A₆. **d** A close-up view of 3′A₆ inside the "pocket" comprised of multiple aromatic/hydrophobic residues (labeled). 3′A₆ also makes four hydrogen bonds (green dashes). For clarity, the 5′ phosphate group of 3′A₆ is omitted. The inset depicts the outline of the pocket around 3′A₆ (in green) and the outline of the molecule 3′A₆ (in gray).

conditions as those produced diffracting co-crystals of RNA3-AA with rA3G_R8/E259A, only RNA3-GA with rA3G_R8/E259A co-crystallization was successful. Failure to form crystals with ssRNA containing the AG and GG dinucleotide motifs under the experimental conditions is likely caused by their weak binding affinities with rA3G_R8/E259A, and, thus, lack of stable complex formation. These results also confirm that rA3G favors the purine dinucleotide sequence motifs AA/GA over AG/GG.

The complex structure of the RNA3-GA with rA3G_R8/E259A was determined as a monomer with one rA3G_R8/E259A molecule binding to one ssRNA molecule at 1.83 Å (Fig. 5a). Six nucleotides were resolved in the GA and rA3G_R8/E259A complex: 5′-U₄G₅A₆U₇U₈U₉-3′. The overall rmsd between the models of the AA and GA complexes is 0.394 over 3052 pairs of atoms, indicating that they are essentially the same structure except 5′A₅ being replaced by 5′G₅. The electron density map at a 1.5 sigma contour level depicts 5′G₅ unambiguously (Fig. 5b, c). While the hydrophobic interaction with the 3′A₆ bound to the pocket is similar (the solvation energy effect of −2.31 kcal/mol), much-reduced hydrophobic interaction was found with 5′G₅, as the solvation energy effect was −0.74 kcal/mol. Further analysis shows that the groove allows the shift of 5′G₅ (vs 5′A₅) so that the reoriented 5′G₅ avoids clashes of its carbonyl at the C-6 position with the F286 main-chain

carbonyl group and uses its N1 to form a hydrogen bond with the F286 main-chain carbonyl (2.8 Å, Fig. 5d). While this shift also avoids the clash of 5′G₅ 2-NH2 with W127, the reoriented 5′G₅ moves ~0.3–0.5 Å away from W127/F126 compared with 5′A₅, weakening the packing interaction of 5′G₅ with W127/F126. Additionally, the shifted 5′G₅ C-6 carbonyl group is still in the proximity of the main-chain carbonyl of P267 (4.0 Å) and in a negative environment at the C-terminal end of an alpha helix. These factors can explain the less favorable binding of 5′G₅ than 5′A₅.

## rA3G bound to dsRNA with AA dinucleotide-containing overhangs

Both 5′- and 3′-overhang dsRNA complex structures display an eight-base pair A-form dsRNA (dsRNA₈). Even though the bulk of two individual dsRNA₈ situated at two different locations on the rA3G surface, the unpaired AA dinucleotides on the 5′- or 3′-overhang of each dsRNA₈ is the common key to be recognized by rA3G (Fig. 6). The molecular contacts between rA3G and the AA dinucleotides observed in both 5′- and 3′-overhang dsRNA structures (Fig. 6a) are essentially identical to those between rA3G and the AA dinucleotide in the complex with ssRNA. Three sharp RNA kinks are present at both 5′- and

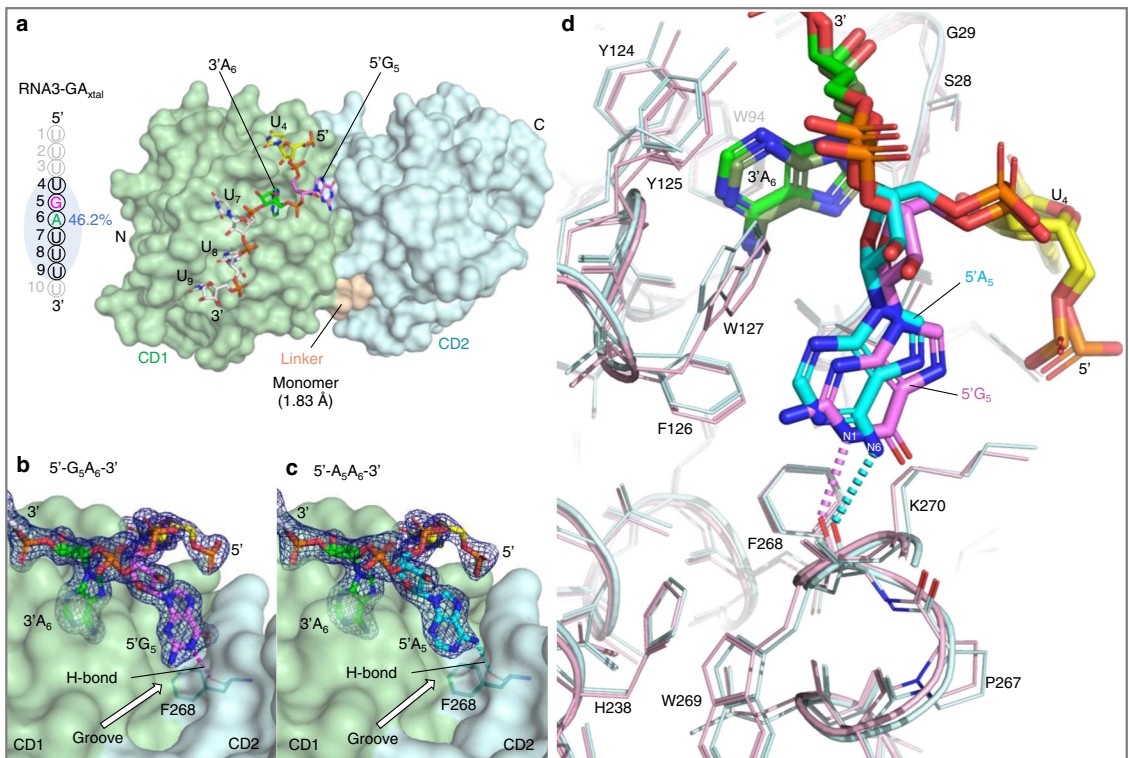

**Fig. 5 | Co-crystal structure of RNA3-GA$_{xtal}$ in complex with rA3G$_{R8/E259A}$ (PDB ID: 8EDJ) and comparison of the sequence motifs 5′-G$_5$A$_6$-3′ and 5′-A$_5$A$_6$-3′ bound to the pocket and the groove on the rA3G surface. a** RNA3-GA$_{xtal}$ and its co-crystal structure with rA3G$_{R8/E259A}$ as a monomer. RNA bases not marked in light gray are resolved. The sequence motif 5′-G$_5$A$_6$-3′ is colored as 5′G$_5$ in pink and 3′A$_6$ in green. The percentage of the buried RNA surface is indicated. The CD1 (pale green), the CD2 (pale cyan), and the linker $_{194}$RHL$_{196}$ (wheat) are labeled. **b** A 2Fo-Fc electron density map of the sequence motif 5′-G$_5$A$_6$-3′, showing 5′G$_5$ (pink) inside the groove and 3′A$_6$ (green) inside the pocket (contoured at 1.5 σ level). The hydrogen bond (H-bond) between N1 of 5′G$_5$ and the main-chain carbonyl group of F268 is shown (in pink). **c** A 2Fo-Fc electron density map of the sequence motif 5′-A$_5$A$_6$-3′, showing 5′A$_5$ (cyan) inside the groove and 3′A$_6$ (green) inside the pocket (contoured at 1.5 σ level). The hydrogen bond (H-bond) between N6 of 5′A$_5$ and the main-chain carbonyl group of F268 is shown (in cyan). **d** Superimposition of the two RNA molecules RNA3-GA$_{xtal}$ and RNA3-AA$_{xtal}$, showing a slight shift in the 5′G$_5$ position and the H-bond between N1 of 5′G$_5$ and the main-chain carbonyl group of F268 (in pink). The H-bond between N6 of 5′A$_5$ and the main-chain carbonyl group of F268 is also shown (in cyan). Their corresponding rA3G molecules are shown as stick representations with rA3G bound to RNA3-GA$_{xtal}$ in pink and rA3G bound to RNA3-AA$_{xtal}$ in cyan.

3′-end of each of the two AA dinucleotides (Fig. 6a inset and Supplementary Fig. 7b), suggesting deformation of RNA occurs to facilitate AA dinucleotide binding to rA3G, or the tight binding of the AA dinucleotide on rA3G provides a strong anchoring point to afford the deformation of RNA on both ends of each A residue to accommodate the path of the ssRNA and dsRNA attached to the protein surface. These sharp RNA kinks also allow rA3G to bind to the unpaired AA dinucleotide located right at a dsRNA fork junction which is analogous to the cases of AA dinucleotides located on the ssRNA stretches connecting two dsRNA secondary structures present in many locations of HIV-1 RNA genome[54].

In the 5′-overhang dsRNA co-crystal structure, dsRNA$_8$ has little contact with rA3G, resulting in a relatively weaker electron density for part of the dsRNA (Fig. 6b). A C-G base pair (C$_5$-G$_{12}$) 3′ to the bound AA core (5′-A$_3$A$_4$-3′) is hovering over the surface of CD1-loop 3 ($_{58}$VYP$_{60}$). The side chain of the loop 3 residue Y59 shows no clear electron density at 1 sigma contour level, suggesting it doesn't make strong contact with the RNA.

In the 3′-overhang dsRNA co-crystal structure, there are multiple contacts between dsRNA$_8$ and rA3G, resulting in a relatively rigid structure with defined electron density for the dsRNA. The dsRNA$_8$ has a G-C base pair (G$_8$-C$_1$) 5′ to the bound AA ssRNA core (5′-A$_9$A$_{10}$-3′) (Fig. 6c), with G$_8$ (in the sequence 5′-G$_8$A$_9$A$_{10}$-3′) making five hydrogen bonds with S28 through its base/sugar moiety (bond lengths 2.5–3.7 Å), including two that are guanine base specific (carbonyl/amyl groups of S28 peptide backbone with N2/N3 groups of G8, respectively). Additional five hydrogen bonds are present between the side

chains of R24/N176/N177 and the phosphate backbone of C$_5$, G$_6$, G$_7$, and G$_8$ of the dsRNA (bond lengths 2.8–3.7 Å). Contacts with the phosphate backbone and the base G$_8$ in the dsRNA region enhances the overall stability of the dsRNA$_8$ structure.

## Effect of mutations on unpaired AA dinucleotide RNA binding, virion packaging, and HIV restriction

We tested the virion packaging and HIV-1 restriction by the wild-type rA3G and hA3G and their mutants using HIV-1 (ΔVif) in cell culture (Fig. 7). These rA3G/hA3G mutations were designed based on the co-crystal structures to disrupt or weaken the interactions with the AA dinucleotide around the aromatic/hydrophobic pocket and the CD1/CD2 interface identified herein (Fig. 7a–c). Mutant 1, 2, and 3 carry mutations on loop 1 ($_{26}$ILS$_{28}$) or loop 7 ($_{124}$YY$_{125}$ and $_{126}$FW$_{127}$). These residues are all critically involved in binding to the AA dinucleotides in the rA3G structures and have been reported by multiple previous biochemistry studies to affect RNA association and dimerization in human A3G[32–34,37,40]. Both mutant 4 and 5 carry two mutations from the CD2 domain, F268A and K270A, intended to disrupt the observed interaction with the 5′A of the AA dinucleotide at the CD1-CD2 interface. Mutant 5 also carries additional CD1 mutations, Y154R/P179E, to disrupt their contacts with CD2 loop 3 as in the RNA-bound conformation. Thus, mutant 5 containing F268A/K270A plus Y154R/P179E mutations is intended to disrupt the binding of the 5′A by CD2 residues and, at the same time, weaken the CD1/CD2 interface. Mutant 6 carries three mutations on CD1 h6, L184E, A187E, and T188E, intended to disrupt the CD1/CD2 interface of the RNA-bound form.

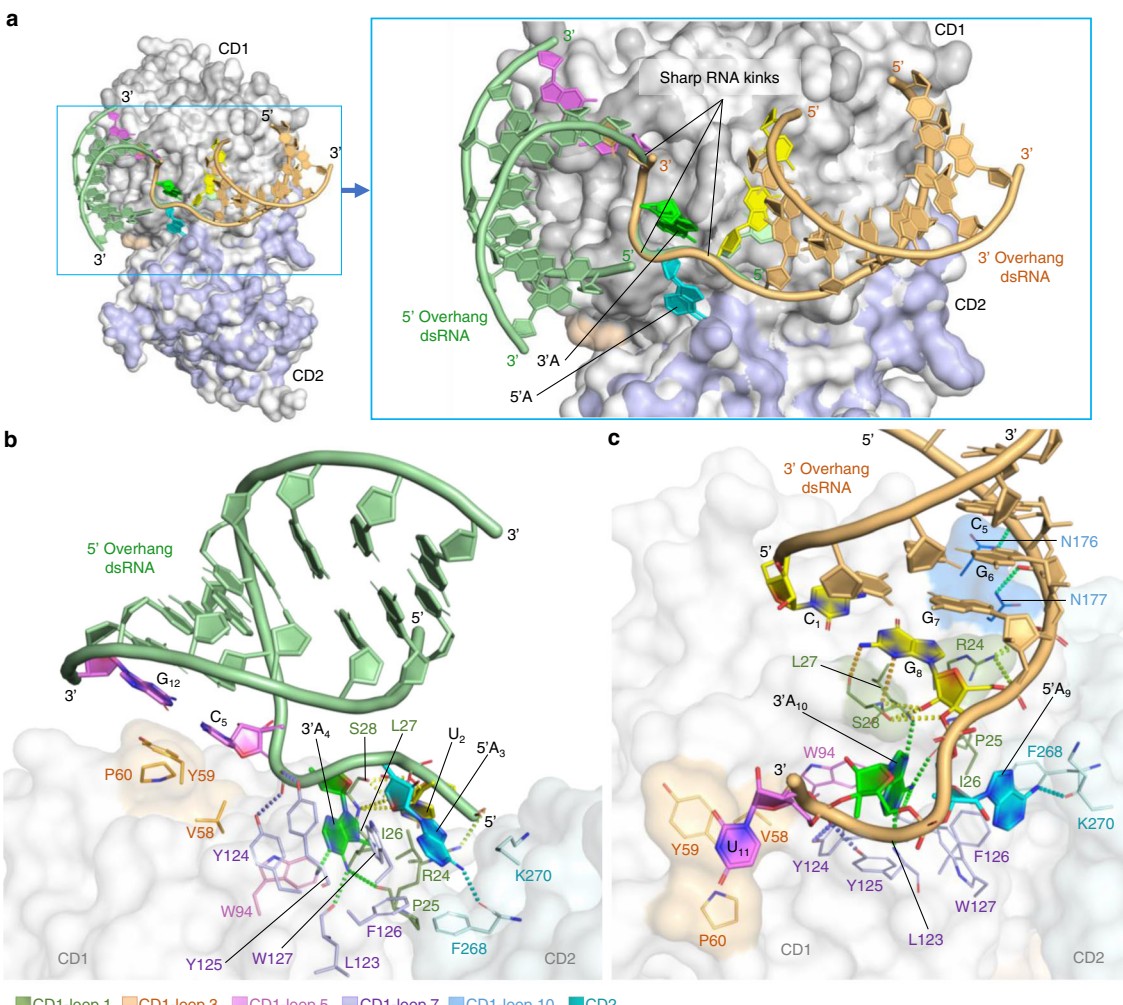

**Fig. 6 | Surface and stick representations of rA3G in complex with 5′- or 3′-overhang dsRNA (PDB ID: 7UU5 and 7UU3).** The color key of the amino acid residues is below panel **b**. **a** Superimposition of 5′-overhang (in pale green) and 3′-overhang dsRNA (in pale orange) bound to their corresponding rA3G molecule, showing overlap of the AA dinucleotide at the rA3G binding site with 5′A (in cyan) and 3′A (in green). The first base pair $G_{12}$-$C_5$ at the junction of the 5′-overhang dsRNA is in violet. The first base pair $C_1$-$G_8$, at the junction of the 3′-overhang dsRNA is in yellow. The three sharp kinks of the phosphate backbone are indicated (also

see Supplementary Fig. 7d). **b** Unique features of the 5′-overhang dsRNA bound to rA3G. The base pair $G_{12}$-$C_5$ (in violet) contacts the surface of rA3G near $_{58}VYP_{60}$ on CD1-loop 3 (orange-colored area). **c** Unique features of the 3′-overhang dsRNA bound to rA3G. The base pair $C_1$-$G_8$ (in yellow) contacts rA3G through $G_8$, forming multiple hydrogen bonds with R24 and S28 (two green-colored areas) and additional hydrogen bonds on the same RNA strand between the phosphate backbone and N176/N177 (blue-colored area).

The results show that virion packaging is most defective in rA3G mutants 1, 2, and 3 (Fig. 7d, e), which are predicted to affect interactions with both As of the AA dinucleotide. The virion packaging is also partially affected in rA3G mutants 4, 5, and 6 (Fig. 7d, e). The restriction of HIV-1 (ΔVif) infectivity was severely impacted in mutant 1, 2, and 3, and partially in mutant 4, 5, and 6 of rA3G (Fig. 7f). Similar results in virion packaging and restriction of HIV-1 (ΔVif) infectivity were obtained with a set of hA3G mutants carrying corresponding mutations as in rA3G (Fig. 7g–i). The only difference is mutant 4 (F268A/K270A) of hA3G, which showed close to the wild-type activity even though its virion packaging is reduced compared with the wild-type hA3G. The effect on packaging and HIV-1 restriction by mutant 4 (F268A/K270A) of rA3G is also the lowest among the six mutants tested. Mutants 5 and 6 show a similar effect on reducing the virion packaging and HIV-1 restriction for rA3G and hA3G.

To examine the RNA binding affinity of mutants 1–6 of rA3G, we performed RNA binding assays comparing the same concentration gradient of purified RNA-free rA3G mutant proteins binding to AA dinucleotide RNA (Fig. 7j and Supplementary Fig. 8). Mutants 1, 2, 3,

and 6 have a much-reduced affinity to AA dinucleotide ssRNA, while mutants 4 and 5 show a modest reduction in the RNA affinity than the wild-type rA3G. These RNA binding results indicate that there is a correlation between disrupted AA dinucleotide RNA binding and virion packaging/HIV-1 restriction activity for rA3G and hA3G. Our results are consistent with the observation by York and colleagues that AA dinucleotides are enriched in the cross-linked nucleotides in the immature HIV virions[31].

## Discussion

In this study, we describe four structures of a soluble primate rA3G variant from rhesus macaque, $rA3G_{R8/E259A}$, in complex with AA and GA dinucleotide-containing RNA molecules in the form of ssRNA and dsRNA with 5′- or 3′-overhangs. These structures reveal a molecular mechanism of rA3G specifically recognizing and binding unpaired 5′-AA-3′ and 5′-GA-3′ dinucleotide sequences. Hydrophobic interactions are the main driving force as $rA3G_{R8/E259A}$ monomer binds 3′A via a highly specific hydrophobic/aromatic cave-like pocket on the non-catalytic domain CD1 and 5′A or 5′G in a hydrophobic groove formed at the CD1-CD2 interface identified in all four RNA-bound rA3G structures

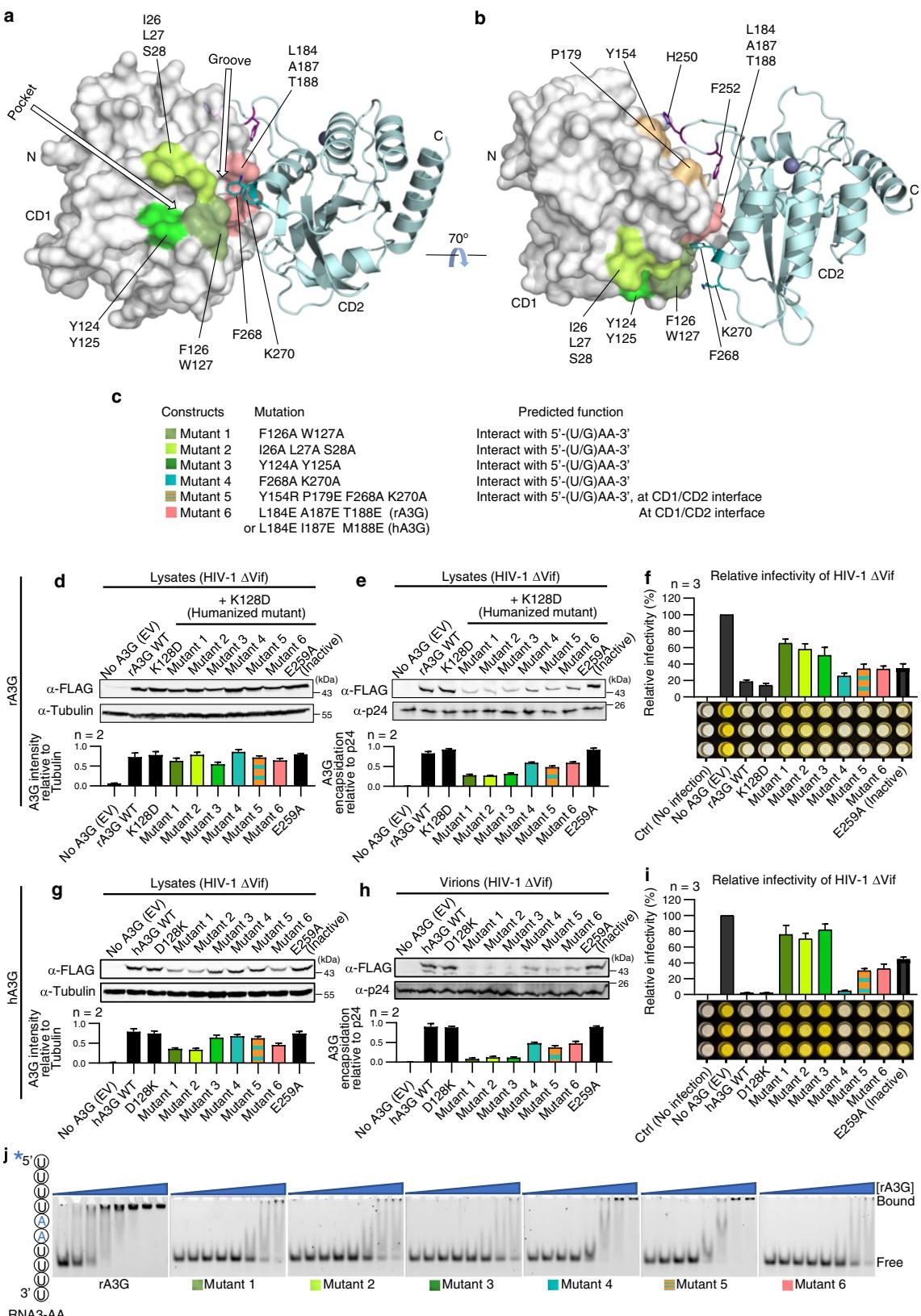

in this study. Comparison of rA3G binding features for the ssRNA AA dinucleotide with that of modeled hA3G structure reveals a set of identical residues interacting with the AA dinucleotide RNA sequence (Supplementary Fig. 9a). A protein sequence alignment of A3G homologs shows well-conserved key residues for AA dinucleotide binding, suggesting that the sequence-specific recognition of AA and

GA RNA are general traits of the RNA binding property of A3G homologs (Supplementary Fig. 9b).

When in complex with 5′- or 3′-overhang dsRNA containing unpaired AA dinucleotide, RNA-mediated rA3G dimers are formed. These RNA-mediated dimers can have different dimerization arrangements (Fig. 2b, e and Supplementary Fig. 10a, b), depending on

**Fig. 7 | Test of virion packaging and antiviral activity of rA3G and hA3G mutants designed to disrupt the binding to AA dinucleotide RNA. a, b** Surface representation of rA3G showing locations of six mutational groups (mutants 1–6). **c** Predicted function of mutants 1–6 for rA3G and hA3G. The mutated residues are identical in rA3G and hA3G except mutant 6, where rA3G has A187E/T188E, and hA3G has I187E/M188E. **d–f** The results of virion packaging of rA3G mutants and their restriction of HIV-1 infectivity using HIV-1(ΔVif). **g–i** The results of virion packaging of hA3G mutants and their restriction of HIV-1 infectivity using HIV-

1(ΔVif). The virion packaging results are quantified as results of duplicate independent experiments ($n = 2$) and the restriction of HIV-1 infectivity assays are presented as results of triplicated independent experiments ($n = 3$). All graphs shown in Fig. 7 are represented as mean values ± SD. **j** Representative gel images of EMSA using the 6-FAM labeled RNA3-AA (10 nM) and the purified wild-type rA3G$_{R8}$ or six rA3G mutant proteins of various concentrations (1.4, 4.1, 12.3, 37, 111, 333, 1000, and 2000 nM). $n = 3$ independent experiments. The asterisk marks the location of 6-FAM. Source data are provided as a Source Data file.

whether the ssRNA is 5′- or 3′-overhang and the length of the dsRNA region. These RNA-mediated rA3G dimer structures are also different from the RNA-free rA3G dimer that is mediated through protein-protein contact via the CD1 h6-h6 interactions (Supplementary Fig. 10c). In the RNA-bound forms, the CD1 h6 of rA3G is completely buried within the interface with CD2, thus excluding the CD1 h6-h6 dimerization observed for apo-rA3G.

Previously reported A3H dimer structures also show dimerization through the bound RNA without any protein-protein contact[50–52]. Comparing the dimer structures of rA3G-RNA and A3H-RNA (PDB ID: 5W3V, 5Z98) reveals additional similarities and differences. In both cases, there are symmetric short ssRNA overhangs on both ends of the dsRNA interacting with the respective proteins, suggesting their critical role in RNA-assisted dimer formation. Moreover, at least one unpaired base in each overhang is captured by a similar aromatic/ hydrophobic pocket on a monomer. However, the polarity of the bound ssRNA is opposite when considering the similar hydrophobic pockets in both rA3G and A3H binding to the ssRNA. Second, rA3G has more extensive interactions with the 3-nt ssRNA that can be at both 5′- or 3′-overhang, with CD2 of rA3G participating in binding to one of the two adenine bases. A3H interacts with 2-nt ssRNA on 5′-overhang. Third is that rA3G mainly interacts with one strand (the AA dinucleotide-containing strand) of the dsRNA, while A3H shows significant contact with both strands of dsRNA.

An extensive body of prior hA3G research has shown the promiscuous nature of RNA binding with a preference for G-rich and A-rich sequenses[31,55]. In this study, our structural and biochemical data explain why rA3G specifically favors unpaired 5′-AA-3′ over 5′-GG-3′ or any other dinucleotides. The 3′A binds to the hydrophobic pocket of rA3G as the major anchoring point, and the 5′A binds to the CD1-CD2 interface groove to enforce the anchoring. Our data also reveal that, while the CD1 pocket can only bind the 3′A, the interface groove for the 5′A can also bind a 5′G with less favorable binding energy but cannot bind 5′C/5′U well. Our structural data, together with our biochemical data, suggest that rA3G has the strongest affinity for RNA containing unpaired 5′-AA-3′ ($K_D$ of 10 to 17 nM), a lesser affinity for 5′-GA-3′ ($K_D$ of 47 nM, ~3 to 4-fold less), much less affinity for 5′-AG-3′ ($K_D$ of 200 nM, ~15 times less), and even lower affinity for 5′-GG-3′ ($K_D$ of 1500 nM).

HIV-1 genomic RNA (and lentiviral RNA in general) is known for having a higher percentage of adenine (A) (36.2%) while having a low cytosine (C) (17.6%)[56]. This bias toward A nucleotide in the ssRNA regions becomes even more profound in the viral RNA structure as determined by SHAPE, reaching 47.5% for unpaired A, while having only 21.3, 19.2, and 11.9% for unpaired U, G, and C, respectively[54,57]. While unpaired Gs are likely associated with viral RNA chaperons[58] such as HIV NC, unpaired AAs are not known to be tightly sequestered by the HIV NC or other viral proteins. Therefore, a preference for unpaired AA sequences in RNA could be advantageous for A3G as a host restriction factor. Viral RNA with multiple numbers of unpaired and accessible AA (and, to a lesser extent, GA) dinucleotides can result in dimerization and multimerization of bound rA3G on RNA. Such "aggregation" of A3G to the enriched AA dinucleotides on viral RNA could serve as a potential mechanism for selective A3G encapsidation into HIV virions. Further study can focus on in vivo validation by reducing or increasing the unpaired AA dinucleotide content in the HIV-1 genome using synonymous mutagenesis and investigating the

impact on A3G HIV-1 virion packaging and HIV restriction. With improved protein sample quality of human and other primates A3G, one can conduct structural studies related to sequence-specific nucleic acid recognition to uncover biochemical properties and functional mechanisms of human and primate A3G proteins, and other double domain APOBECs.

## Methods

### Protein expression and purification

A soluble variant of a primate APOBEC3G from rhesus macaque (rA3G, protein accession code: AGE34493) carries a replacement of N-terminal domain loop 8 ($_{139}$CQKRDGPH$_{146}$ to $_{139}$AEAG$_{142}$, designated as rA3G$_{R8}$) was constructed in the pET-sumo vector (Thermo Fisher) and generated a sumo fusion that carries an N-terminal 6xHis tag and a PreScission protease cleavage site[41]. Briefly, rA3G expressing *E.coli* Rosetta™(DE3)pLysS cells were cultured at 37 °C in LB medium supplemented with 50 μg/ml kanamycin until OD$_{600}$ reached 0.3, the growth temperature was then lowered to 16 °C, and *E. coli* cells were further cultured. When OD$_{600}$ reached 0.7–0.9, protein expression was induced by IPTG. *E. coli* cells were further cultured at 16 °C overnight and harvested by centrifugation (5000×g, 15 min, 4 °C). Cell pellet was lysed in buffer A (25 mM HEPES pH 7.5, 0.5 M NaCl, 20 mM MgCl$_2$, 0.5 mM TCEP, and 60 μg/ml RNase A) by sonication. Cell supernatant was obtained after centrifugation (20,000×g, 60 min, 4 °C) and the 6xHis sumo rA3G$_{R8}$ fusion was captured by Ni-NTA agarose gravity-flow chromatography and eluted with 0.5 M imidazole in buffer B containing 0.25 M NaCl (buffer B: 50 mM HEPES pH 7.5, 0.25 M NaCl, and 0.5 mM TCEP). Ni elution fractions were concentrated and switched to buffer B containing 0.5 M NaCl and 0.1 mM EDTA. RNase A (1 μg/μl) and RNase T1 (2 units/μl) were included in the concentrated Ni elution fraction and left overnight in a cold room. PreScission protease was then added for 2 to 3 h to remove the sumo tag, followed by the initial size-exclusion chromatography (SEC) step using Superdex 200 pg (HiLoad™ 16/600) column equilibrated in buffer B with 0.5 M NaCl on an AKTA Pure chromatography system. The peak fractions of the cleaved rA3G were pooled, concentrated, and passed through a HiTrap 5 ml Heparin HP column using the binding buffer (buffer B with 0.5 M NaCl) and elution buffer (buffer B with 2 M NaCl). The final size-exclusion chromatography polishing step was done on a Superdex 200 pg (HiLoad™ 16/600) column equilibrated in buffer B with 0.25 M NaCl. Peak fractions were pooled, concentrated to 7–9 mg/ml, and stored at −80 °C until use. Sequences of all mutant constructs were verified by Sanger sequencing. Mutant proteins were purified using the same protocol. The final polishing step by SEC was omitted if the absorbance ratio (A$_{260}$/A$_{280}$) of the protein sample after heparin column purification was less than 0.6, indicating the protein sample was largely free of RNA. Each protein sample was quantified by NanoDrop with the absorption coefficient correction and verified by 12% SDS-PAGE (Supplementary Fig. 8).

A host codon optimized HIV nucleocapsid DNA fragment (NC, protein accession code: NP_579881) was synthesized (Thermo Fisher) and cloned into an *Escherichia coli* expression vector pGEX6P1 as an N-terminal glutathione *S*-transferase and nucleocapsid fusion (GST-NC). The sequence of the GST-NC construct was verified by Sanger sequencing. Expression of GST-NC or GST in the *E. coli* XA90 strain was carried out following the expression protocol for rA3G[41]. Cell pellet of

GST-NC or GST was lysed in buffer A. Cell supernatant of GST-NC or GST was obtained after centrifugation (20,000×*g*, 60 min, 4 °C). GST-NC or GST was captured by Glutathione agarose gravity-flow chromatography and eluted with 40 mM reduced Glutathione in buffer B containing 0.5 M NaCl. Elution fractions were concentrated, and a similar RNase treatment as described above was carried out with the GST-NC sample. The initial size-exclusion chromatography (SEC) step for GST-NC or GST was carried out using Superdex 200 pg (HiLoad™ 16/600) column equilibrated in buffer B with 0.5 M NaCl. The peak fractions of GST were pooled, concentrated, and stored at −80 °C. The peak fractions of GST-NC were pooled, concentrated, and switched to buffer B with 0.25 M NaCl. It was further purified with HiTrap 5 ml Heparin HP column using binding buffer (buffer B with 0.25 M NaCl) and elution buffer (buffer B with 2 M NaCl). Peak fractions were pooled and concentrated. The final size-exclusion chromatography polishing step was carried out on Superdex 200 Increase 10/300 GL column equilibrated in buffer B with 0.25 M NaCl. Peak fractions were pooled, concentrated to 4−5 mg/ml, and stored in −80 °C until use. The absorbance ratio ($A_{260}/A_{280}$) of the purified GST-NC was 0.53 and the purified GST was 0.51, as measured by NanoDrop. Purified GST-NC and GST were verified by 12% SDS-PAGE (Supplementary Fig. 2a).

### RNA preparation and native gel electrophoresis

RNA oligonucleotides were purchased from Integrated DNA Technologies (IDT) and the sequences are listed in Supplementary Table 1. A stock solution of 1 mM was prepared for single-stranded RNA (ssRNA) substrates using water. A stock solution of 2 mM was prepared for RNA oligonucleotides intended for making 5′- or 3′-overhang dsRNA in the annealing buffer (60 mM KCl, 6 mM Tris-HCl, pH 8, and 0.2 mM $MgCl_2$). dsRNA with overhangs was prepared by mixing an equal volume of the two RNA strands and annealed by heating to 95 °C for 5 min, followed by slow cooling to room temperature.

Formation of annealed dsRNA with overhangs using FAM labeled RNA or unlabeled RNA was verified by 20% native gel electrophoresis (acrylamide:bis-acrylamide ratio of 19:1) in 0.5X TBE buffer (40 mM Tris base, 45 mM boric acid, 1 mM EDTA) (Supplementary Fig. 1). SYBR™ Gold Nucleic Acid Gel Stain (Thermo Fisher) was used for staining unlabeled RNA (Supplementary Fig. 1b). Amersham™ Typhoon™ Biomolecular Imager (GE Healthcare) was used to visualize gel images.

### Crystallization and structure determination

rA3G$_{R8/E259A}$ carrying the inactive mutation E259A (with or without K128D) mutation was purified using the same protocol as described above. rA3G-RNA complex was prepared by mixing rA3G (4 mg/ml) with RNase inhibitor and then added to the ssRNA with a molar ratio 1 to 1 (rA3G$_{R8/E259A}$:RNA3-AA$_{xtal}$), or to the annealed dsRNA with overhangs with molar ratio 2 to 1 (rA3G:RNA1-AA$_{xtal}$ or rA3G:RNA2-AAxtal). After incubation on ice for 1 h, precipitation was removed by centrifugation (21,000×*g*, 2 min, 4 °C). Initial screening was carried out by sitting-drop vapor diffusion method using ARI Crystal Gryphon Robot (ARI) and crystallization screening kits (Qiagen and Hampton Research) at 18 °C. Crystallization hits were optimized by hanging drop vapor diffusion method at 18 °C. The diffraction-quality crystals of RNA1-AA$_{xtal}$ complex were obtained with the reservoir solution consisting of 1.8 M Na/K phosphate pH 6.9; crystals of RNA2-AA$_{xtal}$ complex were obtained with the reservoir solution consisting of 0.1 M KCl, 0.1 M Tris-HCl pH 8, and 12% PEG 2000 monomethyl ether; crystals of RNA3-AA$_{xtal}$ complex were obtained with the reservoir solution consisting of 0.22 M $Na_2SO_4$ and 20% PEG 3350.

Co-crystallization trials of rA3G$_{R8/E259A}$ and RNA3-NN$_{xtal}$ where NN = AA, GA, AG, or GG were carried out in parallel by hanging drop vapor diffusion method at 18 °C. Guided by the crystallization conditions that produced diffracting co-crystals of rA3G$_{R8/E259A}$ and RNA3-AA$_{xtal}$, we used a three-by-three screening grid with a reservoir solution combination of 0.18/0.20/0.22 M [$Na_2SO_4$] and 18/20/22% [PEG 3350].

Diffraction-quality crystals of rA3G$_{R8/E259A}$ and RNA3-GA$_{xtal}$ complex were obtained with the reservoir solution comprised of 0.18 M $Na_2SO_4$ and 20% PEG 3350, while no crystal was observed with RNA3-AG and RNA3-GG. We carried out another crystallization screening for RNA3-AG/rA3G and RNA3-GG/rA3G using the PACT suite (Qiagen) with 96 defined chemical solutions. One hit was identified, but it turned out to be the same as the hit for the apo-rA3G.

Crystals were transferred to the synthetic mother liquor supplemented with suitable amounts of glycerol for cryoprotection and flash-cooled in liquid $N_2$. X-ray diffraction data were collected at the Advanced Photo Source (GM/CA@APS, Argonne National Laboratory) beamlines 23ID-B and 23ID-D, and at the Advanced Light Source (ALS, Lawrence Berkeley Laboratory) beamline 5.0.3. Data were processed and scaled using HKL-2000 or automated data processing in JBluIce. Initial phase information was obtained by molecular replacement method with PHENIX using the crystal structures of the rA3G N-terminal domain (PDB 5K81) and the hA3G catalytic domain (PDB 3IR2) as search models. The structural model was refined using PHENIX[59] and modified with COOT[60]. RNA was built manually in COOT and further refined using ERRASER[61]. Data collection statistics and refinement parameters are summarized in Table 1. Electrostatic surface potentials were calculated with APBS[62,63]. Buried surface areas were calculated with QtPISA[64–66]. Structure images were prepared with PyMOL (The PyMOL Molecular Graphics System, Version 2.5.3 Schrödinger, LLC.).

### Electrophoretic mobility shift assay (EMSA)

RNA with 6-FAM at 10 nM was titrated by rA3G in 20 μl reaction volume containing 50 mM HEPES pH 7.5, 250 mM NaCl, 1 mM DTT, 0.4 units/μl RNase inhibitor, and 10% glycerol. Reaction mixtures were incubated on ice for 10 min and analyzed by 8% native PAGE at 4 °C. An acrylamide:bis-acrylamide ratio of 29:1 was used in preparing 8% native gel for overhang dsRNA complexes, and a higher acrylamide: bis-acrylamide ratio of 72.5:1 was for ssRNA complexes. Amersham™ Typhoon™ Biomolecular Imager (GE Healthcare) was used to visualize gel images. ImageQuant TL (v8.1, GE Healthcare) was used for image quantification. Dissociation constant $K_D$ was calculated using GraphPad Prism version 8.4.3 for Windows, GraphPad Software, San Diego, California USA, www.graphpad.com. Three independent experiments were carried out for each RNA molecule.

### RNA binding assay

RNA binding was assayed using gel filtration. Samples of rA3G protein alone (1.7 or 3.3 μM), RNA alone (1.7 or 3.3 μM), or rA3G-RNA mixture at 1:1 molar ratio, were applied to a Superdex 200 Increase 10/300 GL column (Cytiva) in buffer B with 0.25 M NaCl and 0.4 units/μl RNase inhibitor. Elution profiles of relevant samples were superimposed, and peak positions were compared.

### Single-cycle replication assay

The Single-cycle replication assays with the VSV-G pseudotyped HIV-1 LAI ΔVif ΔEnv were performed[41]. HEK293T cells were cultured in DMEM supplemented with 10% FBS and Penicillin (100 U/mL)/Streptomycin (100 μg/mL) and maintained at 37 °C, 5% $CO_2$. The cells were co-transfected with 500 ng of pHIV-1 LAI ΔVif ΔEnv (NIH AIDS Reagent Program), 180 ng of pMD2.G (VSV-G, Addgene), and between 60−200 ng of pcDNA3-N-terminus-1xFLAG tagged A3G expressing plasmid to achieve similar expression levels of A3G wild-type and mutants, using Lipofectamine™ 3000 Transfection Reagent (Thermo Fisher). Empty pcDNA3 was used to equalize the amount of transfected plasmid DNA. The medium was changed after 24 h and virus-containing supernatant was collected 24 h after the medium change. The supernatant was filtered through a 0.45 μm PVDF filter (Millipore). To determine viral infectivity, TZM-B1 reporter cells (NIH AIDS Reagent Program) were plated at $1 × 10^4$ cells per well of a 96-well plate and

infected with virus normalized by p24 levels (QuickTiter Lentivirus Titer Kit, Cell Biolabs Inc). After 48 h of infection, the cells were washed with PBS, and the levels of infectivity were measured by colorimetric detection with a Pierce™ β-galactosidase assay reagent (Thermo Scientific) using a spectrophotometer (absorbance at 405 nm, SpectraMax® ID5). The relative viral infectivity was quantitatively compared under the condition of No A3G as 100%.

## Quantitative immunoblotting

Filtered virions from the Single-cycle assays were precipitated with NaCl (0.3 M final) and PEG-6000 (8.5% final) at 4 °C for 6 h and centrifuged at 3724×$g$ at 4 °C for 30 min. The transfected HEK293T cells (cell lysates) and the precipitated virions were lysed using 1x RIPA buffer (Sigma). Western blot analysis was performed with anti-FLAG M2 mAB (F3165, Sigma, 1:3000) to determine A3G levels in lysates and virions, respectively. For each internal loading control, α-tubulin (GT114, GeneTex, 1:5000) was used for cell lysates and p24 (Cat #3537, NIH AIDS Reagent Program, 1:3000) was used for virions. Cy3-labeled goat-anti-mouse mAb (PA43009, GE Healthcare, 1:3000) was subsequently used as a secondary antibody. Cy3 signals were visualized using Typhoon RGB Biomolecular Imager (GE Healthcare) and quantitatively measured with ImageQuant TL (GE Healthcare) to show normalized A3G levels.

## Reporting summary

Further information on research design is available in the Nature Portfolio Reporting Summary linked to this article.

## Data availability

The data supporting the findings of this study are available within the paper and its Supplementary Information files, and also available from the corresponding author upon request. Atomic coordinates and structure factors have been deposited in the PDB database under accession codes 7UU5(rA3G$_{R8/E259A}$/RNA1-AA$_{xtal}$), 7UU3 (rA3G$_{R8/E259A}$/RNA2-AA$_{xtal}$), 7UU4 (rA3G$_{R8/E259A}$/RNA3-AA$_{xtal}$), and 8EDJ (rA3G$_{R8/E259A}$/RNA3-GA$_{xtal}$). The atomic models used in this study are available in the PDB database under accession codes 5K81, 3IR2, 6P40, 6P3X, and 6WMA. Source data are provided with this paper.

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

## Acknowledgements

This work is supported by the NIH grant R01 AI150524 to X.S.C. Beamlines of GM/CA@APS have been funded by the National Cancer Institute (ACB-12002) and the National Institute of General Medical Sciences (AGM-12006, P30GM138396). The Eiger 16 M detector at GM/CA-XSD was funded by NIH grant S10 OD012289. The ALS-ENABLE beamlines are supported in part by the National Institutes of Health, National Institute of General Medical Sciences, grant P30 GM124169. The Pilatus detector on 5.0.1. was funded under NIH grant S10OD021832. We are grateful to L. Chelico for providing pHIV-1 LAI ΔVif ΔEnv vector and to HIV AIDS Reagent Program for providing HIV-1 p24 monoclonal antibody.

## Author contributions

X.S.C., H.Y., and K.K. planned the experiments. H.Y. prepared recombinant proteins, performed biochemical assays, carried out crystallographic experiments, and determined the structures (except the co-crystal structure of rA3G$_{R8/E259A}$/RNA3-GA$_{xtal}$). S.L. performed model refinement. K.K. performed antiviral activity experiments and HIV genome analysis. J.P. prepared the recombinant protein rA3G$_{R8/E259A}$,

performed crystallographic experiments with ssRNA containing purine dinucleotide motifs AA, GA, AG, and GG, determined and refined the co-crystal structure of RNA3-GA$_{xtal}$ bound to rA3G$_{R8/E259A}$. All authors analyzed the data. H.Y., X.S.C., and K.K. wrote the manuscript. All authors edited the manuscript.

## Competing interests

The authors declare no competing interests.
