## [Peer Review File · Nature Communications]

Structural basis of sequence-specific RNA recognition by the antiviral factor APOBEC3GREVIEWER COMMENTS

Reviewer #1 (Remarks to the Author):

In this manuscript, Yang et al. determined the co-crystal structures of a cytidine deaminase, rhesus macaque APOBEC3G (rA3G), bound to three types of RNA. They report structural mechanisms to understand how A3G interacts with RNA for virion packaging.

The authors first performed EMSA to find optimal RNA binders with high affinity to rA3G. Using three types of RNA that have adenine dinucleotides (AA), the authors successfully determined three co-crystal structures. They have different configurations and modes for their interaction although unpaired AA in RNAs is commonly key to be recognized by rA3G: an aromatic/hydrophobic cage of the rA3G CD1 and a cavity conformed by the CD1 and CD2 are important. Finally, the authors carried out mutational analysis of the rA3G interfaces for RNA recognition and tested the effects of interface mutations on rA3G (and human A3G) expression, virion packaging and infectivity against vif-deficient HIV-1. The results support their proposed structural mechanisms of rA3G-RNA interaction, which are based on their determined structures.

Twenty years passed since A3G was identified as an anti-retroviral enzyme targeting into viral particles although we have not yet known about structural mechanisms about how A3G interacts with RNA for virion packaging. This is largely due to some technical difficulties to solubilize A3G protein and to find optimal RNA binders for co-crystalization. Therefore, the authors' achievement by Yang et al. should be appreciated in the field of virology and structure biology. I believe that additional data and a few minor corrections are warranted to improve the manuscript as below.

Major comments:

1. In the manuscript, the authors should describe why they initially focused on particular dsRNA and nucleotides containing different dinucleotides to find optimal RNA binders.
2. As the authors cite, it has been reported that A3G preferentially binds to A-rich nucleotides as well as G-rich ones (York et al. PLOS pathogens 2016 doi:10.1371/journal.ppat.1005833 and Bogerd et al. RNA 2008 doi:10.1261/rna.964708). The authors should examine (for example, by in silico calculation) and discuss whether GG dinucleotide (or GA) can be accommodated into the cave and cavity that were determined in this study.
3. There are observed three different domain orientations in the rA3G CD1-CD2 apo structures. Are there any potential effects of crystal-packing contacts in the determined structures. If so, please describe in the manuscript.

Minor comments:

p9, line 11; "Supplementary Fig. 8" need to be changed to "Supplementary Fig. 6".

p9, line 13; "IDT" should be "Integrated DNA Technologies".

Reviewer #2 (Remarks to the Author):

This manuscript describes the structures of RNA molecules, especially those contain single-stranded AA dinucleotide, in complexes with a mutant rhesus macaque A3G. This is a straightforward study with clear results illuminating how A3G selectively binds A-rich RNA to facilitate its packaging into the virion. I have relatively minor comments.

1. The model building and refinement of the structures need to be improved. The structures have high percentage of Ramachandran outliers and poor RNA backbone geometry (especially for 7UU5 and 7UU3). The PDB validation report shows that the 7UU5 crystal is twinned, which is not mentioned in the manuscript. This should be treated appropriately.
2. Please provide the sequences of the duplex regions of the RNA used in crystallization.
3. Figure 1a-c: please label the discrete bands of rA3GR8 bound to the AA dinucleotide RNA mentioned in the main text, as they are not that obvious. The numbers of significant figures for the K_d values are not appropriate (the precision is excessive; the errors are much larger). This is a common issue for some other later figures and Table one too (e.g. resolution "3.004", completeness "0.9824", and not consistent throughout the table).
4. In the SEC binding analysis, no interaction was detected between rA3G and none-AA dinucleotides, but the EMSA assay indicated they still had nanomolar K_d 's?
5. Figure 2c: please label the loop 10-h6 of CD1 near the zinc-catalytic center of CD2 of rA3G that interacts with the RNA interaction core.
6. Figure 2d-e: label CD1 & CD2
7. Page 5: does replacement of A6 in ssDNA to G,C,U also reduce the ssDNA binding to rA3G?
8. Page 7: Is it correct that the HIV-1 (DelVif) infectivity was severely impacted in mutant 1, 2, and 3, and partially in mutant 4, 5, and 6 of rA3G (Fig. 4f)? Should it be "the restriction of" HIV-1 infectivity was severely impacted in mutant 1,2,3, and partially in mutant 4, 5, and 6 of rA3G?

Reviewer #3 (Remarks to the Author):

Description: In this manuscript, the authors determined the structure of a rhesus macaque A3G in complex with three RNA molecules with single-stranded AA dinucleotides either adjacent to a dsRNA at the 5' end or the 3' end, or in unstructured ssRNA. It was found that the rA3G specifically binds to the 3'A in the AA dinucleotide using an aromatic/hydrophobic cage formed by residues in CD1, the non-catalytic domain of rA3G, while the 5'A in AA dinucleotides was bound between a new interface between CD1 and CD2. The rA3G forms RNA-assisted dimers without any apparent protein-protein

interactions. It was suggested that binding to the unpaired AA dinucleotides plays an essential role in rA3G virion incorporation and thus anti-HIV activity.

General critique: These studies describe the structure of A3G in complex with RNA, an important interaction that is critical for the antiviral activity of A3G proteins, since binding to RNA is essential for its packaging into virions and hence for its antiviral activity. Whether APOBEC3 proteins specifically interact with viral RNA has long been the subject of debate, and this study provides some insight into the mechanism, and suggests that unpaired AA dinucleotide binding is responsible for virion incorporation, and perhaps the A-rich nature of HIV-1 genomic RNA provides some enhancement of A3G encapsidation.

Despite the strengths of the manuscript, evidence that unpaired AA dinucleotides play a role in virion incorporation is lacking. Supplementary Figure 7 suggests that there is a correlation between A3G binding and AA dinucleotides as determined by CLIP-seq data. However, the data correlated AA dinucleotides with peaks of A3G binding in a 100-nt window, which is not relevant, since the A3G crosslinks and the AA dinucleotides could be potentially 100-nt away from each other. The CLIP-seq data should show precisely which nucleotides in the HIV genome are involved in the cross-links. The authors should determine whether AA dinucleotides are enriched in the cross-linked nucleotides. An experiment that would provide convincing data would be to mutate some of the AA dinucleotides in HIV-1 sequences and show that the peaks in the cross-linking map generated by CLIP-seq are diminished. These results directly test the main conclusion of the study that AA dinucleotides are involved in RNA binding in the context of a virion in a physiologically relevant condition.

Another concern is that the 3'A in the AA dinucleotides was not replaced with G, C, or U to determine the relative rA3G binding by EMSA. In Supplementary Figure 4b, this analysis was performed for the 5'A in AA dinucleotides and it was observed that AA is indeed a preferred binding site followed by GA, and then UA or CA. Although Figure 1a, 1b, and 1c determined that the AA dinucleotide is preferred over GG, CC or UU dinucleotides, it is not clear whether the A at the 3' position will be preferred and to what extent, when the 5' A nucleotide is maintained. EMSA analysis of 5'AA should be compared to 5'GA, 5'CA, and 5'UA to determine their relative efficiency of binding to rA3G.

Specific suggestions:

I found the structure images throughout the paper rather small and it was difficult to see the points the authors were making in the results section. I had to enlarge the images and print them in an enlarged format in order to follow the points made in the text. I suggest that the size of the figures should be enlarged so that the reader can actually see the points the authors are referring to in the text.

We thank the reviewers for their insightful comments on our manuscript. We have added new structural data regarding the binding of rA3G to purine dinucleotide -GA- containing RNA, new biochemical data on rA3G binding to -AG-, -AC, and -AU-, and RNA binding property of HIV nucleocapsid NC. We have revised the manuscript based on reviewers' comments.

Point-by-point response:

REVIEWER COMMENTS

Reviewer #1 (Remarks to the Author):

In this manuscript, Yang et al. determined the co-crystal structures of a cytidine deaminase, rhesus macaque APOBEC3G (rA3G), bound to three types of RNA. They report structural mechanisms to understand how A3G interacts with RNA for virion packaging.

The authors first performed EMSA to find optimal RNA binders with high affinity to rA3G. Using three types of RNA that have adenine dinucleotides (AA), the authors successfully determined three co-crystal structures. They have different configurations and modes for their interaction although unpaired AA in RNAs is commonly key to be recognized by rA3G: an aromatic/hydrophobic cage of the rA3G CD1 and a cavity conformed by the CD1 and CD2 are important. Finally, the authors carried out mutational analysis of the rA3G interfaces for RNA recognition and tested the effects of interface mutations on rA3G (and human A3G) expression, virion packaging and infectivity against vif-deficient HIV-1. The results support their proposed structural mechanisms of rA3G-RNA interaction, which are based on their determined structures.

Twenty years passed since A3G was identified as an anti-retroviral enzyme targeting into viral particles although we have not yet known about structural mechanisms about how A3G interacts with RNA for virion packaging. This is largely due to some technical difficulties to solubilize A3G protein and to find optimal RNA binders for co-crystallization. Therefore, the authors' achievement by Yang et al. should be appreciated in the field of virology and structure biology. I believe that additional data and a few minor corrections are warranted to improve the manuscript as below.

Major comments:

1. In the manuscript, the authors should describe why they initially focused on particular dsRNA and nucleotides containing different dinucleotides to find optimal RNA binders.

We have now included this information as requested (p3 line 20):

“Preliminary RNA binding tests using an RNA molecule comprised of a dsRNA with both 5' and 3' overhangs yielded a pronounced peak shift on size-exclusion chromatography (SEC). As a result, three types of RNA were designed to evaluate sequence specific RNA binding: dsRNA with a short 5' or 3' overhang (RNA1 or RNA2, Fig. 1a) and ssRNA (RNA3, Fig. 1a). Two variable bases NN (NN = GG, AA, CC, or UU) were introduced to the short 5' and 3' overhangs at the dsRNA junction, and in the middle of the short ssRNA sequences.”

2. As the authors cite, it has been reported that A3G preferentially binds to A-rich nucleotides as well as G-rich ones (York et al. PLOS pathogens 2016 doi:10.1371/journal.ppat.1005833 and Bogerd et al. RNA 2008 doi:10.1261/rna.964708). The authors should examine (for example, by in silico calculation) and discuss whether GG dinucleotide (or GA) can be accommodated into the cave and cavity that were determined in this study.

We have now analyzed the structures and confirmed our previous conclusion that the cave-like pocket on CD1 is adenosine (A) specific (for the 3'A of the 5'-AA-3' dinucleotide, p6 line 11), while the interface groove between CD1 and CD2 can accommodate both adenosine A and guanosine G (for the 5'A/G of the 5'-AA-3' or 5'-GA-3') with its preference for A (see boxed section below for details, also p6, line 21). Therefore, 5'-GA-3' can be accommodated into the interface groove and the cave-like pocket (with the 5'G in groove and the 3'A in

pocket). However, for 5'-GG-3', even though the 5'G can fit into the interface groove, the 3'G will not be able to bind into the cave-like pocket as a G will have clash inside the cave-like pocket on CD1.

Consistent with the structural analysis, the dissociation constant K_D of ssRNA containing 5'-AA-3' is 10 nM (Fig. 1a), 5'-GA-3' is 47 nM (Fig. 1b), 5'-AG-3' is 200 nM (Fig. 1b, new data during revision), and 5'-GG-3' is 1.5 μ M (Fig. 1a) measured by EMSA. Please also see relevant discussions in **Response to point #7 Reviewer 2**.

Structural analysis

The cave-like pocket (on CD1)

Panels a,b: a 3'A fits in the cave-like pocket. Local surface electrostatic potential around N6 of 3'A is negative and N6 forms two H-bonds with two main-chain carbonyl groups from P25/L123.

Panels c, d: a 3'G doesn't fit in the cave-like pocket. Local surface electrostatic potential around O6 of 3'G is negative, rendering it unfavorable for 3'G Binding. The carbonyl group (O6) of 3'G is highly unfavorable when in close proximity (3.0-3.1Å) to the two carbonyl groups from P25/L123 (a). There is also a clash between N2 of 3'G with the aromatic ring of Y125.

The interface groove (at the interface between CD1 and CD2)

Panels e, f: a 5'A fits in the interface groove. Local surface electrostatic potential around N6 of 5'A is negative and N6 forms one H-bond with the main-chain carbonyl group of F268.

Panels g,h: a 5'G can also fit but less favorable than an A. Local surface electrostatic potential around O6 of 5'G carbonyl group is negative, rendering it unfavorable for 5'G binding, as O6 carbonyl group is too close to the main-chain carbonyl group of F268, with two other carbonyl groups nearby on both sides of F268 (P267/W269) to create a negative potential environment around the negatively charged 5'G carbonyl.

We also investigated the purine selectivity by rA3G using crystallization method and obtained a structure of rA3G with 5'-GA-3' containing RNA to 1.83 Å resolution. We have included this new data in the revised manuscript (p7 line 1). The structure shows that while the interactions of the cave-like pocket with the 3'A is the same as before, much reduced hydrophobic interaction was found between the interface groove and the 5'G, as the solvation energy effect was -0.74 kcal/mol (vs. -3.07 kcal/mol with 5'A). Further analysis shows that the groove allows the shift of the 5'G so that F286 mainchain carbonyl makes H-bond with 5'G N1 and avoid clashes with 5'G C-6 carbonyl group, but with the penalty of increasing the distance with two aromatic sidechains and having the 5'G C6-carbonyl group near the carbonyl-rich environment at the C-terminus of an alpha helix that is considered to be negatively charged. The interface groove, while can accommodate A or G, does not fit a pyrimidine (C/U) well. This new structure, together with the initial three structures and biochemical data, delineates the molecular mechanism of selectivity of A at the cave-like pocket and a preference of A over G at the interface groove.

Additionally, we investigated the RNA binding property of HIV-NC using the same set of short RNA molecules (p4 line 1; Fig 1c). Our results show that the best binders for HIV-NC are RNA containing GG motifs, which is consistent with literature that HIV-NC favors unpaired Gs. HIV-NC has similar weak affinity to AA/CC/UU, which suggests that preference of unpaired AA sequences in RNA could be advantageous for A3G as a host restriction factor, as AA would not be tightly sequestered by HIV NC or other known viral proteins. We have included this new data in the revised manuscript.

3. There are observed three different domain orientations in the rA3G CD1-CD2 apo structures. Are there any potential effects of crystal-packing contacts in the determined structures. If so, please describe in the manuscript.

This is a good point. When different conformations of a protein are observed, there is always a possibility of crystal packing effect. This study shows that hydrophobic interaction is the main driving force for rA3G and RNA association. In the apo forms, the hydrophobic RNA binding surfaces are available, and they have tendency to associate with each other (or with other proteins) during purification and crystallization processes. The three apo structures of rA3G show different crystal packing, and show different CD1-CD2 orientation from each other, but the differences between the three apo structures are much less pronounced when compared with their difference with the RNA-bound structures. Importantly no CD1-CD2 conformational differences are observed for the four rA3G-RNA complex structures despite of different space group and crystal packing and the RNA types in the complex. We believe that, for the RNA-bound structures the effect of crystal packing could be excluded. We have added this explanation in this revision on p5 line 20.

Minor comments:

p9, line 11; "Supplementary Fig. 8" need to be changed to "Supplementary Fig. 6".

Yes, this has now been corrected with the correct figure number in the revised manuscript (p10 line 37).

p9, line 13; "IDT" should be "Integrated DNA Technologies".

Yes, this has now been corrected (p11 line 14).

Reviewer #2 (Remarks to the Author):

This manuscript describes the structures of RNA molecules, especially those contain single-stranded AA dinucleotide, in complexes with a mutant rhesus macaque A3G. This is a straightforward study with clear results illuminating how A3G selectively binds A-rich RNA to facilitate its packaging into the virion. I have relatively minor comments.

1. The model building and refinement of the structures need to be improved. The structures have high percentage of Ramachandran outliers and poor RNA backbone geometry (especially for 7UU5 and 7UU3). The PDB validation report shows that the 7UU5 crystal is twinned, which is not mentioned in the manuscript. This should be treated appropriately.

We further refined all the structures (7UU3, 7UU4, 7UU5). Now the Ramachandran plots are improved. Also, RNA backbone geometry is greatly improved after we used ERRASER (Chou, 2013 Nature Methods 10: 74 – 76) to refine RNA (See the updated PDB validation reports).

Thank you for pointing out the twin character of 7UU5 crystal. Sorry, we didn't realize that before. The refinement was carried out without using "twin law". Now we have the opportunity looking into this. After the refinement with "twin law", we got 0.1880/0.2398 for the R/Rfree. Compared with the R/Rfree (0.1891/0.2406) from refinement without using "twin law", it is not a big difference. Also, the map is not significant improved. This indicates that there is very minor twinning for this crystal, and thus, we think the refinement without the "twin law" is more reasonable.

2. Please provide the sequences of the duplex regions of the RNA used in crystallization.

We have now included the sequences of the duplex regions of the RNA used in crystallization (p4 lines 22, 24).

3. Figure 1a-c: please label the discrete bands of rA3GR8 bound to the AA dinucleotide RNA mentioned in the main text, as they are not that obvious. The numbers of significant figures for the Kd values are not appropriate

(the precision is excessive; the errors are much larger). This is a common issue for some other later figures and Table one too (e.g. resolution “3.004”, completeness “0.9824”, and not consistent throughout the table). We acknowledge that this was indeed the case that the discrete bands were not obvious in the case with ssRNA. Therefore, we added labels to RNA1-AA and RNA2-AA (revised Fig 1). The main text has now been updated to reflect this change (p3, line 29).

We acknowledge that the precision is excessive for the number of significant figures in the manuscript. We have now corrected them throughout the manuscript.

4. In the SEC binding analysis, no interaction was detected between rA3G and none-AA dinucleotides, but the EMSA assay indicated they still had nanomolar K_D 's?

During SEC binding analysis, sample dilution is inevitable since diffusion occurs as the small volume of sample is mixed with the much larger elution volume of buffers and passes through the relatively long path of the column. In addition, dissociated RNA is effectively separated from rA3G preventing reassociation. As a result, when using rA3G-RNA complex at 1.7 to 3.3 μ M starting concentration, peak shifts were only observed with AA dinucleotide RNA substrates. Another factor may be the complex of rA3G binding to AA containing RNA is more stable that can survive the SEC process much better than other complexes. We didn't explore higher complex concentrations in order to conserve our experimental material as SEC analysis uses relatively large amounts of purified protein and RNA.

5. Figure 2c: please label the loop 10–h6 of CD1 near the zinc-catalytic center of CD2 of rA3G that interacts with the RNA interaction core.

We have revised accordingly (revised Fig. 3b).

6. Figure 2d-e: label CD1 & CD2

We have revised Figure 2d-e accordingly (revised Fig. 4a, 4c)

7. Page 5: does replacement of A6 in ssDNA to G,C,U also reduce the ssDNA binding to rA3G?

We measured the K_D for AG/AC/AU during revision. Yes, replacement of A₆ in ssDNA to G, C, or U also reduces the ssRNA binding to rA3G.

	K_D (nM)
-AA- 5' UUUUA ₅ A ₆ UUUU	10
Pair 1 5' UUUUG ₅ A ₆ UUUU	47
5' UUUUG ₅ G ₆ UUUU	1500
Pair 2 5' UUUUC ₅ A ₆ UUUU	253
5' UUUUC ₅ C ₆ UUUU	ND (affinity too low to be determined)
Pair 3 5' UUUUU ₅ A ₆ UUUU	124
5' UUUUU ₅ U ₆ UUUU	ND (affinity too low to be determined)
5' UUUU ₄ A ₅ G ₆ UUUU	204
5' UUUU ₄ A ₅ C ₆ UUUU	339
5' UUUU ₄ A ₅ U ₆ UUUU	199

As we have communicated in the reply to Reviewer #1 question 2, we also investigated the purine selectivity by rA3G using crystallization method and obtained co-crystals and solved a structure of rA3G with 5'-GA-3'

ssRNA (5' UUUUG₅A₆UUUU) under the similar conditions as with the 5'-AA-3' containing ssRNA but failed to obtain co-crystal of rA3G with ssRNA containing 5'-AG-3' or 5'-GG-3'. We have included this new structure of rA3G with 5'-GA-3' ssRNA in the revised manuscript (p7 line 1). Please also see the reply to Reviewer #1 question 2.

8. Page 7: Is it correct that the HIV-1 (DelVif) infectivity was severely impacted in mutant 1, 2, and 3, and partially in mutant 4, 5, and 6 of rA3G (Fig. 4f)? Should it be “the restriction of” HIV-1 infectivity was severely impacted in mutant 1,2,3, and partially in mutant 4, 5, and 6 of rA3G?

The Reviewer#2 is correct that the restriction of HIV-1 infectivity was severely impacted in mutant 1,2,3, and partially in mutant 4, 5, and 6 of rA3G. We have now corrected this sentence in the revision (p8 line 27).

Reviewer #3 (Remarks to the Author):

Description: In this manuscript, the authors determined the structure of a rhesus macaque A3G in complex with three RNA molecules with single-stranded AA dinucleotides either adjacent to a dsRNA at the 5' end or the 3' end, or in unstructured ssRNA. It was found that the rA3G specifically binds to the 3'A in the AA dinucleotide using an aromatic/hydrophobic cage formed by residues in CD1, the non-catalytic domain of rA3G, while the 5'A in AA dinucleotides was bound between a new interface between CD1 and CD2. The rA3G forms RNA-assisted dimers without any apparent protein-protein interactions. It was suggested that binding to the unpaired AA dinucleotides plays an essential role in rA3G virion incorporation and thus anti-HIV activity.

General critique: These studies describe the structure of A3G in complex with RNA, an important interaction that is critical for the antiviral activity of A3G proteins, since binding to RNA is essential for its packaging into virions and hence for its antiviral activity. Whether APOBEC3 proteins specifically interact with viral RNA has long been the subject of debate, and this study provides some insight into the mechanism, and suggests that unpaired AA dinucleotide binding is responsible for virion incorporation, and perhaps the A-rich nature of HIV-1 genomic RNA provides some enhancement of A3G encapsidation.

(1). Despite the strengths of the manuscript, evidence that unpaired AA dinucleotides play a role in virion incorporation is lacking. Supplementary Figure 7 suggests that there is a correlation between A3G binding and AA dinucleotides as determined by CLIP-seq data. However, the data correlated AA dinucleotides with peaks of A3G binding in a 100-nt window, which is not relevant, since the A3G crosslinks and the AA dinucleotides could be potentially 100-nt away from each other. The CLIP-seq data should show precisely which nucleotides in the HIV genome are involved in the cross-links. The authors should determine whether AA dinucleotides are enriched in the cross-linked nucleotides. An experiment that would provide convincing data would be to mutate some of the AA dinucleotides in HIV-1 sequences and show that the peaks in the cross-linking map generated by CLIP-seq are diminished. These results directly test the main conclusion of the study that AA dinucleotides are involved in RNA binding in the context of a virion in a physiologically relevant condition.

Reviewer#3 has raised the following four points.

1) The main conclusion of the study

We acknowledge that the reported results did not offer the evidence “AA dinucleotides are involved in RNA binding in the context of a virion in a physiologically relevant condition”, as raised by reviewer#3. We have revised the Discussion section accordingly to improve the clarity of this important aspect of the manuscript (p10 line 7).

2) Precise correlation between peaks of A3G binding as determined by CLIP-seq data and AA dinucleotides.

Since the precise coordinates of the nucleotides in the cross-linked peaks/valleys from York and colleagues' study (PLoS Pathogens, 12, e1005833, 2016) are not available to the public, we are not able to provide the

precise correlation. Therefore, we decided to remove Supplementary Figure 7. Instead, we have cited the results of short A3G recognition motifs determined by York *et. al.* in the same study. See details in the following paragraph.

3) Determine whether AA dinucleotides are enriched in the cross-linked nucleotides

York and colleagues (PLoS Pathogens, 12, e1005833, 2016) have determined the short sequence motifs (in the form of deoxynucleotide sequence) as most frequently present in RNA fragments crosslinked to A3G (via Thio-U) shown in their Figure 5B. A screenshot of a partial Figure 5B showing the A3G motifs is shown below. AA dinucleotides are indeed enriched under their immature virion condition (the important step for the recruitment of A3G into the progeny virions), which is consistent with our observation that rA3G favors unpaired AA dinucleotide sequences. It's still unclear to us what causes the pattern change at different physiological stages before and after infection, which requires further study.

We have cited this in our revised manuscript (p9 line 39).

4) Mutate some of the AA dinucleotides in HIV-1 sequences and check whether the peaks in the cross-linking map generated by CLIP-seq are diminished.

We agree that these experiments are critical to validate whether AA dinucleotides are involved in RNA binding in the context of a virion in a physiologically relevant condition. This is one of the directions for the further in-depth study to fulfill our understanding.

As A3G competes with HIV-NC in binding to HIV RNA during virion packaging process, we thought to analyze the RNA binding property of HIV-NC using the same set of short RNA molecules. Our results show that the best binders for HIV-NC are RNA containing GG motifs, which is consistent with literature that HIV-NC favors unpaired Gs. HIV-NC has similar weak affinity to AA/CC/UU, which suggests that preference of unpaired AA sequences in RNA could be advantageous for A3G as a host restriction factor, as AA would not be tightly sequestered by viral chaperons. We have included this new data in the revised manuscript (p4 line1, Fig 1c).

(2). Another concern is that the 3'A in the AA dinucleotides was not replaced with G, C, or U to determine the relative rA3G binding by EMSA. In Supplementary Figure 4b, this analysis was performed for the 5'A in AA dinucleotides and it was observed that AA is indeed a preferred binding site followed by GA, and then UA or CA. Although Figure 1a, 1b, and 1c determined that the AA dinucleotide is preferred over GG, CC or UU dinucleotides, it is not clear whether the A at the 3' position will be preferred and to what extent, when the 5' A nucleotide is maintained. EMSA analysis of 5'AA should be compared to 5'GA, 5'CA, and 5'UA to determine their relative efficiency of binding to rA3G.

We measured the K_D for AG/AC/AU during revision. Yes, replacement of A_6 in ssDNA to G,C,U also reduces the ssRNA binding to rA3G.

	K_D (nM)
5' UUUUA ₅ A ₆ UUUU	10
5' UUUU ₄ A ₅ G ₆ UUUU	204
5' UUUU ₄ A ₅ C ₆ UUUU	339
5' UUUU ₄ A ₅ U ₆ UUUU	199
5' UUUUG ₅ A ₆ UUUU	47
5' UUUUG ₅ G ₆ UUUU	1500

As we have communicated in the reply to Reviewer #1 question 2, we also used crystallography to validate this, and found that rA3G can only be co-crystallized with ssRNA with AA or GA sequences: 5' UUUUA₅A₆UUUU and 5' UUUUG₅A₆UUUU, but not with other ssRNA sequences. We solved the rA3G in complex with the ssRNA containing GA sequence to 1.83 Å, which reveals (together with the AA ssRNA structure) the mechanism of selectivity for AA and GA dinucleotides by rA3G. We have included this new data in the revised manuscript (p7 line 1). Please see the reply to Reviewer #1 question 2.

Specific suggestions:

I found the structure images throughout the paper rather small and it was difficult to see the points the authors were making in the results section. I had to enlarge the images and print them in an enlarged format in order to follow the points made in the text. I suggest that the size of the figures should be enlarged so that the reader can actually see the points the authors are referring to in the text.

Thank you for pointing this out. We have revised the size/labels of Fig.2, Fig.4, and revised the labels in Fig. 3 and Fig. 6 to improve the clarity of our figures.

REVIEWERS' COMMENTS

Reviewer #1 (Remarks to the Author):

The authors have addressed each of my comments from the initial review. I believe the manuscript has nicely improved and will be appreciated in the fields of virology and structure biology.

Reviewer #2 (Remarks to the Author):

The authors have thoroughly responded to the comments from the reviewers and the manuscript is much improved. It is suitable for publication now.

Reviewer #3 (Remarks to the Author):

The authors have resubmitted a much-improved manuscript, which includes a structure of rhesus A3G with 5'GA3' containing RNA at a high resolution. These additional data and the structural analysis included in response to reviewer 1 provide a mechanistic explanation for why 5'AA3' is preferred over 5'GA3'. Additionally, the authors have removed the problematic Supplementary Figure 7, suggesting (I believe erroneously) that there is a correlation between A3G binding and AA dinucleotides in CLIP-seq assays. I have no further concerns about this strong structural study that addresses important unresolved questions regarding the specificity of interaction between APOBEC3 proteins and RNA.

REVIEWERS' COMMENTS

Reviewer #1 (Remarks to the Author):

The authors have addressed each of my comments from the initial review. I believe the manuscript has nicely improved and will be appreciated in the fields of virology and structure biology.

Reviewer #2 (Remarks to the Author):

The authors have thoroughly responded to the comments from the reviewers and the manuscript is much improved. It is suitable for publication now.

Reviewer #3 (Remarks to the Author):

The authors have resubmitted a much-improved manuscript, which includes a structure of rhesus A3G with 5'GA3' containing RNA at a high resolution. These additional data and the structural analysis included in response to reviewer 1 provide a mechanistic explanation for why 5'AA3' is preferred over 5'GA3'. Additionally, the authors have removed the problematic Supplementary Figure 7, suggesting (I believe erroneously) that there is a correlation between A3G binding and AA dinucleotides in CLIP-seq assays. I have no further concerns about this strong structural study that addresses important unresolved questions regarding the specificity of interaction between APOBEC3 proteins and RNA.

We would like to express our great appreciation to all three reviewers for their constructive comments that helped to improve our manuscript.